# Hedgehog signaling via its ligand DHH acts as cell fate determinant during skeletal muscle regeneration

Alessandra M. Norris[1], Ambili Bai Appu[1], Connor D. Johnson[1], Lylybell Y. Zhou[1], David W. McKellar [2], Marie-Ange Renault [3], David Hammers[1], Benjamin D. Cosgrove [2] & Daniel Kopinke [1] ✉

Successful muscle regeneration relies on the interplay of multiple cell populations. However, the signals required for this coordinated intercellular crosstalk remain largely unknown. Here, we describe how the Hedgehog (Hh) signaling pathway controls the fate of fibro/adipogenic progenitors (FAPs), the cellular origin of intramuscular fat (IMAT) and fibrotic scar tissue. Using conditional mutagenesis and pharmacological Hh modulators in vivo and in vitro, we identify DHH as the key ligand that acts as a potent adipogenic brake by preventing the adipogenic differentiation of FAPs. Hh signaling also impacts muscle regeneration, albeit indirectly through induction of myogenic factors in FAPs. Our results also indicate that ectopic and sustained Hh activation forces FAPs to adopt a fibrogenic fate resulting in widespread fibrosis. In this work, we reveal crucial post-developmental functions of Hh signaling in balancing tissue regeneration and fatty fibrosis. Moreover, they provide the exciting possibility that mis-regulation of the Hh pathway with age and disease could be a major driver of pathological IMAT formation.

Coordinating intercellular signaling between adult stem cells and their niche to balance differentiation and self-renewal is critical for regenerating complex tissues after injury. Successful regeneration of skeletal muscle depends on the interplay of two distinct stem/progenitor cell populations: dedicated muscle stem cells (MuSCs) and fibro/adipogenic progenitors (FAPs). MuSCs differentiate into myoblasts and either fuse together to form new muscle fibers or fuse with existing fibers to repair damaged ones[1–4]. FAPs are a multipotent mesenchymal stem/stromal cell population present in most adult organs including skeletal muscle[5,6]. FAPs build and maintain the extracellular matrix and are crucial during the repair of damaged tissues by secreting beneficial factors[7–15]. With age and disease, FAPs can also differentiate into myofibroblasts, the cellular origin of tissue fibrosis[7,15], and adipocytes, which will form intramuscular fat (IMAT)[7,8,14–19]. The infiltration and replacement of healthy muscle tissue with IMAT and scar tissue, also called fatty fibrosis, is a prominent feature of Duchenne Muscular

Dystrophy (DMD) and other neuromuscular diseases, as well as sarcopenia, obesity, and diabetes[18,20–29]. It remains unclear what signal(s) balance the beneficial functions and multiple fates of FAPs.

Through screens to identify factors that might control FAPs, we recently reported that FAPs are the main ciliated cell type in muscle that can sense and transduce Hedgehog (Hh) signaling[8]. Primary cilia are small cellular antennae that receive and interpret extracellular cues including the Hh pathway[30,31]. Hh is a long-range signaling pathway that is activated through secreted ligands such as Sonic (Shh), Indian (Ihh) or Desert (Dhh) Hedgehog. Uniquely, vertebrate Hh signaling fully relies on the primary cilium for its function[30,31]. In the absence of Hh ligands, the cilium processes the GLI (glioma-associated oncogene homolog) transcription factors into their repressor form to maintain repression of Hh target genes[30,31]. Upon binding of a ligand to the Hh receptor Patched1 (Ptch1), a negative regulator of the pathway, Smoothened (Smo) accumulates in the cilium to activate the Glis[30,31].

[1]Department of Pharmacology and Therapeutics, Myology Institute, University of Florida, Gainesville, FL, USA. [2]Meinig School of Biomedical Engineering, Cornell University, Ithaca, NY, USA. [3]Biology of Cardiovascular Diseases, INSERM, University of Bordeaux, Pessac, France. ✉e-mail: dkopinke@ufl.edu

As a consequence of this intimate relationship between cilia and Hh, only ciliated cells can respond to Hh ligands.

During skeletal muscle development, Hh signaling helps to initiate the myogenic program[32–34]. In mature muscle, while the Hh pathway only displays low activity under homeostatic conditions, acute injuries robustly activate Hh signaling[8,35–38]. This increase in Hh activation during an acute injury indicates that the Hh pathway is required for muscle to regenerate. Contrastingly, in animal models where muscle regeneration is compromised, such as aged mice, the *mdx* mouse (a mouse model of DMD) and in a chronic injury model induced via glycerol, Hh activity is severely blunted[8,35–37,39]. Our recent work demonstrated ectopic reactivation of the Hh pathway during a chronic injury potently inhibited the in vivo differentiation of FAPs into adipocytes via upregulation of the secreted anti-adipogenic factor TIMP3 (Tissue inhibitor of Metalloproteinases 3)[8]. At the same time, we also found that FAP-specific Hh activation promoted myofiber regeneration post injury and prevented the decline of muscle atrophy in *mdx* mice. Thus, these findings establish that Hh is sufficient to block IMAT and promote muscle regeneration. However, whether endogenous Hh signaling is also necessary for muscle regeneration is still unclear. In addition, the identity of the key Hh ligand, and its producing cell type, in skeletal muscle, remains controversial with evidence existing for both Shh[35–37,40] and Dhh[8,38].

Here, we identify DHH as the key Hh ligand through which endogenous Hh balances muscle regeneration and IMAT formation. We find that Hh, via DHH, acts as an endogenous adipogenic brake by inducing TIMP3 and, once lost, results in increased IMAT formation after an acute injury. We also establish that Hh controls myogenesis indirectly through FAPs by inducing the myogenic factor GDF10. Moreover, we define the time window when Hh activity is required to impact adipogenesis and myogenesis. Surprisingly, our data also indicate that ectopic activation of Hh specifically within FAPs pushes FAPs away from an adipogenic towards a fibrogenic fate resulting in massive scar tissue formation and impaired myogenesis. Thus, DHH-induced Hh signaling, through cell autonomous and non-autonomous roles, balances IMAT formation and muscle regeneration pointing to ciliary Hh signaling as a promising therapeutic strategy to combat IMAT formation.

## Results

### Acute muscle injury induces Hedgehog activation, via its ligand DHH, to repress IMAT formation

The identity of the endogenous Hh ligand, and its producing cell type, in skeletal muscle remains unresolved. To objectively determine which Hh ligand is expressed by which cell type during muscle regeneration, we utilized single-cell gene expression analysis. This technique overcomes the restrictions and limits of the previously used method, RT-qPCR of bulk RNA samples, and, at the same time, allows to visualize whole muscle transcriptomics at a cellular resolution. Thus, we quarried unpublished and public single-cell RNAseq (scRNAseq) data sets of skeletal muscle at different time points pre- and post- acute muscle injury, as previously described[41], to ask which Hh ligand is expressed by which cell type (Fig. 1a). Intriguingly, our data reveal that, *Shh* and *Ihh* are undetectable, while *Dhh* is potently expressed within endothelial and neural (mainly comprised of Schwann cells) populations (Fig. 1b). Our data also demonstrate that *Dhh* is rapidly induced upon injury before returning to baseline levels by 10 days post injury (dpi) (Fig. 1b). These data confirm our previous observations that *Dhh* and not *Shh* is the key ligand induced by injury and expressed by Schwann cells within the peripheral nervous system[38,42]. Interestingly, they also point to endothelial cells as another source of DHH similar to what has been reported post ischemic injuries[43].

To determine not only the endogenous function of Hh signaling in general, but that of DHH specifically, we utilized a murine Dhh null mouse model (*Dhh*[−/−]) to assess any defects during muscle regeneration. *Dhh*[−/−] mice are viable, phenotypically normal and have a normal life span[38,44]. Confirming and extending these reports, we find that uninjured 7-month-old *Dhh*[−/−] mice display no gross phenotypical abnormalities including no differences in total body weight, or uninjured Tibialis Anterior (TA) weight compared to littermate controls (*Dhh*[+/−] and *Dhh*[+/+]) (Supplementary Fig. 1a). We first investigated whether DHH induces Hh activation after an acute muscle injury. We selected cardiotoxin (CTX) as our injury model as it induces Hh activity[8] and causes little fat formation[45–47]. Control (*Dhh*[+/+] and *Dhh*[+/−]) and *Dhh*[−/−] mice were injured with CTX and 7 days post injury (dpi). RT-qPCR of whole muscle lysate confirmed complete loss of *Dhh* expression in *Dhh*[−/−] mice (Supplementary Fig. 1b). Lack of DHH resulted in decreased Hh activity as shown by expression for the two Hh downstream targets *Gli1* and *Ptch1* in *Dhh*[−/−] compared to control mice (Fig. 1c). Thus, loss of DHH prevents the endogenous activation of Hh signaling after an acute injury.

To determine whether the lack of Hh activation could impact IMAT formation, we quantified the number of PERILIPIN[+] adipocytes in uninjured and injured TAs from *Dhh*[−/−] and control mice. We found no difference in IMAT formation in uninjured TAs between control and *Dhh*[−/−] mice demonstrating that loss of *Dhh* does not cause ectopic IMAT in the absence of injury (Fig. 1c). Excitingly, there was a significant increase in IMAT 21 days post CTX injury in *Dhh*[−/−] mice compared to controls (Fig. 1c). This observation was independent of sex, and we did not detect any differences between *Dhh*[+/+] and *Dhh*[+/−] control animals (Supplementary Fig. 1c). Besides increased IMAT, most neuromuscular diseases are also affected by widespread tissue fibrosis[28,48,49]. Therefore, we also asked whether loss of *Dhh* would impact injury-induced fibrosis. To visualize the extracellular matrix (ECM), we stained control and *Dhh*[−/−] mice 21 days post CTX injury with picrosirius red, which identifies collagen fibers[11,50]. The lack of difference in the amount of collagen deposition between genotypes indicates that loss of Hh signaling has no effect on fibrosis (Supplementary Fig. 1d). Together, these results demonstrate that DHH acts as a potent adipogenic brake to limit IMAT formation, but not fibrosis, during muscle regeneration.

To corroborate our findings, we also utilized a pharmacological approach to acutely inhibit Hh activation at various time points post injury. The small molecule Gant61 is a selective inhibitor of Gli1- and Gli2- mediated transcription[51,52], while Vismodegib is an FDA-approved selective inhibitor of Smo[53]. Thus, both small molecules prevent endogenous Hh activation downstream of ligand activation. We injured TA muscles of wild type mice with CTX and administered Gant61 at 0 and 2 dpi, while a separate cohort was given Vismodegib daily from 0–4 dpi (Fig. 1d). Successful inhibition of CTX-induced Hh activation after both Gant61 and Vismodegib treatment was determined by RT-qPCR for *Gli1* (Fig. 1d). As a result of repressed Hh signaling, treatment with Gant61 and Vismodegib allowed ectopic IMAT formation (Fig. 1d) similar to that observed after loss of DHH (Fig. 1c). Thus, acute induction of Hh signaling is necessary to limit IMAT formation during muscle regeneration.

To define the cellular mechanism on how DHH represses IMAT formation, we focused on FAPs, the cellular origin of IMAT. Control and *Dhh*[−/−] mice were injured with CTX and TAs were harvested at 3- and 5 dpi. Bromodeoxyuridine (BrdU) was administered 2 h before harvesting to visualize proliferating FAPs (Fig. 1e). We found no difference in total number of PDGFRα[+] FAPs (Fig. 1e), nor was there any difference in FAP proliferation (Fig. 1e) indicating that DHH is not required for FAP expansion. Next, we determined if DHH might impact the differentiation of FAPs. For this, we purified PDGFRα[+] FAPs using Magnetic activated cell sorting (MACS) 3 days after a CTX injury and evaluated the expression of the key early adipogenic genes *Cebpα* (CCAAT enhancer binding protein alpha) and *Ppary* (Peroxisome proliferator activated receptor gamma) (Fig. 1f). Both adipogenic genes were more strongly induced in FAPs from *Dhh*[−/−] mice compared

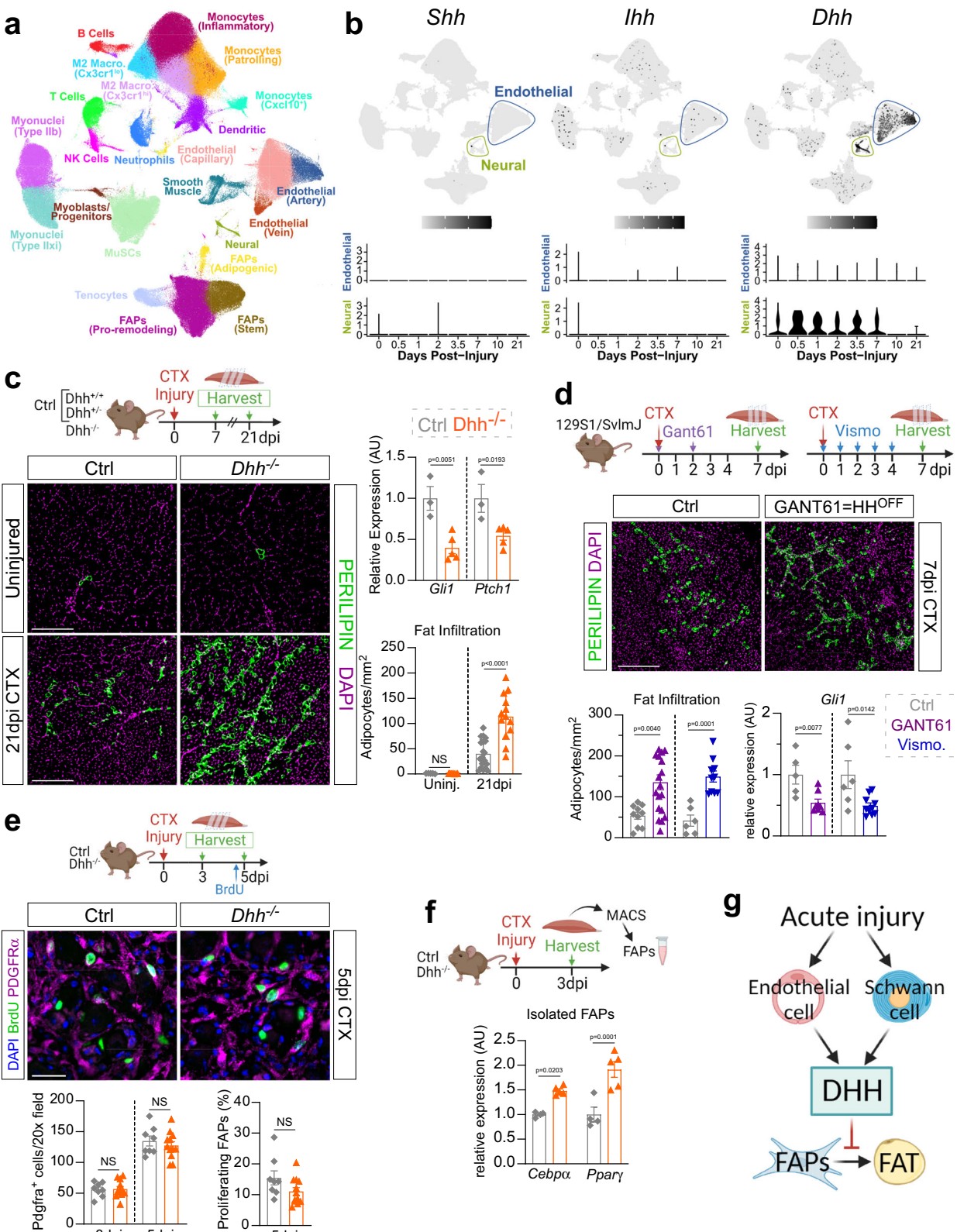

to controls (Fig. 1f) demonstrating that loss of *Dhh* allows the adipogenic differentiation of FAPs. Taken together, an acute muscle injury induces expression of *Dhh* by endothelial and Schwann cells, which in turn blocks the differentiation of FAPs into adipocytes (Fig. 1g).

## DHH is necessary for myofiber regeneration

We and others have shown that ectopic activation of the Hh pathway is sufficient to promote myofiber regeneration[8,37,54,55]. To explore whether endogenous Hh signaling, via its ligand DHH, is also necessary for regenerative myogenesis, we assessed myofiber regeneration 21 days post CTX injury in *Dhh*[−/−] and control mice (Fig. 2a). For this, we measured the cross-sectional area (CSA) of myofibers, a critical metric to evaluate the ability of myofibers to recover from injury[46]. Myofibers were visualized by staining tissue sections for LAMININ, an ECM component, while regenerated fibers were selected based on the presence of centrally located nuclei.

**Fig. 1 | The key Hh ligand DHH acts as an endogenous adipogenic brake during muscle regeneration. a** UMAP (Uniform Manifold Approximation and Projection) plot of 111 published single-cell and single-nucleus RNA-sequencing datasets of skeletal muscle before and after injury. **b** UMAP plots showing aggregated log-normalized expression of the Hh ligands *Shh (Sonic), Ihh (Indian),* and *Dhh (Desert),* and their log-normalized expression within the endothelial or neural cluster at different time points post injury. **c** RT-qPCR for *Gli1* and *Ptch1* at 7dpi (days post injury) post CTX (cardiotoxin) injury (ctrl (control) $n = 3$; *Dhh*$^{-/-}$ $n = 5$). Immuno-fluorescence for adipocytes (PERILIPIN$^+$, green) in uninjured and injured TA (tibialis anterior) muscle. DAPI (purple) marks nuclei. Scale bar: 250 μm. Quantifications of the number of adipocytes per area (Uninjured ctrl $n = 5$, *Dhh*$^{-/-}$ $n = 7$; 21dpi ctrl $n = 20$; *Dhh*$^{-/-}$ $n = 13$). **d** Immunofluorescence for PERILIPIN$^+$ adipocytes (green) at 7 dpi. DAPI (purple) marks nuclei. Scale bar: 250 μm. Quantification of adipocytes per injured area 7 dpi CTX injury of mice administered with Gant61 (vehicle treated:

$n = 10$; Gant61 treated: $n = 18$) or Vismodegib (Vehicle ctrl $n = 6$; Vismodegib treated: $n = 10$). RT-qPCR for *Gli1* of Gant61 (Vehicle ctrl $n = 5$ T; Gant61 treated: $n = 7$) and Vismodegib (vehicle ctrl $n = 6$; Vismodegib treated: $n = 10$) treated mice. **e** Immunofluorescence of BrdU$^+$ cells (green) and FAPs (PDGFRα$^+$ cells; magenta) at 3 dpi. DAPI (purple) marks nuclei. Scale bar: 25 μm. Quantifications of (*Left*) total PDGFRα$^+$ cells per 20x field 3- and 5 dpi in *Dhh*$^{-/-}$ (3dpi $n = 12$; 5dpi $n = 12$) and ctrl (3dpi $n = 8$; 5dpi $n = 8$) mice. (*Right*) Percent of proliferating FAPs (BrdU$^+$ PDGFRα$^+$ cells; % of total FAPs) 5 dpi in *Dhh*$^{-/-}$ ($n = 12$) and ctrls ($n = 8$). **f** RT-qPCR for *Cebpα* and *Pparγ* from MACS-purified FAPs 3 dpi in *Dhh*$^{-/-}$ ($n = 5$ mice) and ctrls ($n = 4$ mice). **g** *Model:* After an acute injury, endothelial and Schwann cells express *Dhh*. DHH-mediated Hh activation represses IMAT formation through inhibiting the adipogenic differentiation of FAPs. All data are represented as mean ± SEM. Each graph data point in c-e represents one TA muscle. An unpaired two-tailed t test or a one-way ANOVA followed by a Dunnet's multiple comparison was used.

After segmenting and measuring myofibers using our recently developed myofiber segmentation pipeline[46], myofibers were false color-coded based on fiber size (Fig. 2a). Similar to IMAT, we found no difference in myofiber CSA in uninjured TAs between *Dhh*$^{-/-}$ and control mice arguing that DHH is dispensable for embryonic myogenesis and adult homeostasis (Fig. 2a). In contrast, myofiber CSA was significantly decreased 21 days post CTX injury in *Dhh*$^{-/-}$ mice compared to controls (Fig. 2a). This reduction in average CSA is due to a change in distribution of fiber size, where *Dhh*$^{-/-}$ mice display a shift from larger to smaller fibers (Fig. 2a). Thus, loss of Dhh impairs myofiber regeneration. As an independent confirmation, we also treated wild type mice with the Hh antagonist Gant61 (Supplementary Fig. 2a) and found a decrease in myofiber CSA 7 days post CTX injury (Supplementary Fig. 2b) similar to loss of *Dhh*. Thus, endogenous Hh activation, via DHH, is critical for myofiber regeneration after an acute injury.

We also injured *Dhh*$^{-/-}$ mice with glycerol (GLY), a highly adipogenic injury[8,45–47,56,57]. First, we compared Hh activation between CTX- and GLY-injured mice via RT-qPCR for *Gli1* and *Ptch1*. Compared to CTX, and similar to our previous data[8], GLY fails to induce Hh signaling (Supplementary Fig. 2c). Importantly, loss of *Dhh* during a GLY injury did not further decrease Hh activity (Supplementary Fig. 2c). Next, we analyzed the number of PERILIPIN-expressing adipocytes as well as myofiber size 21 days post GLY injury and found no differences between *Dhh*$^{-/-}$ and control mice (Supplementary Fig. 2d, e). These findings indicate that loss of *Dhh* post GLY injury has no impact on Hh activity levels, thereby not causing any additional increase in IMAT. Thus, the differential activation of Hh signaling depends on the type of injury and the levels of Hh activity dictate the amount of IMAT allowed to form.

Next, we sought to explore how Hh, via DHH, affects the early steps of regenerative myogenesis. For this, we crossed a global tamoxifen-dependent Cre deleter with a recently generated Dhh floxed allele[43,58] (CAGGCre-ER$^{TM}$ x Dhh$^{c/c}$, called Dhh cKO) (Fig. 2b). This approach allows for widespread and acute loss of *Dhh* upon tamoxifen administration right before an injury insult. Using RT-qPCR, we confirmed successful loss of *Dhh* expression and reduced Hh activity (by *Gli1* expression) in Dhh cKO mice 7 days post CTX injury compared to control littermate mice (Supplementary Fig. 2f, g), similar to *Dhh*$^{-/-}$ mice (Fig. 1c). Importantly, inducible deletion of *Dhh* led to a decrease in myofiber CSA 7 days post CTX injury (Supplementary Fig. 2h) phenocopying the global *Dhh* null phenotype (Fig. 2a). To investigate the role of DHH on early myogenesis, after TMX administration and washout, TAs were injured with CTX and harvested at 3- and 5 dpi (Fig. 2b). At 3 days post CTX injury, we already observed a dramatic reduction in MYOG-expressing myoblasts (Fig. 2c). Fittingly, this resulted in a strong reduction in newly formed MYH3$^+$ (embryonic myosin heavy chain 3) myofibers at 5 dpi. Thus, loss of Hh activity results in a reduced myoblast pool thereby impairing early myogenesis and subsequent myofiber repair (Fig. 2d).

## Ectopic Hedgehog activation impairs both adipogenesis and myogenesis

To determine the timing for when Hh activation is required during the regenerative process, we pharmacologically activated the Hh pathway at different time points post CTX or GLY injury via administration of the Smoothened Agonist, SAG[59] (Fig. 3a). We first evaluated the kinetics of SAG by administering a single dose 2 days post GLY injury and assessed Hh induction through the expression of *Gli1* and *Ptch1* 6- and 12 h post injection (Supplementary Fig. 3a). Interestingly, we find that SAG rapidly induces Hh signaling resulting in a transcriptional response within 6–12 h post injection (Supplementary Fig. 3a). To achieve sustained Hh activity, we administered SAG at days 0-, 2- and 4 post injury. Temporal activation was carried out by a single SAG dose either on the day of injury (0 dpi), 2 dpi, or 4 dpi (Fig. 3a). We confirmed Hh activation by measuring *Gli1* expression 7 days after injury via RT-qPCR. As expected, we observed the highest Hh activation in both injury models when SAG was administered repetitively every other day (Fig. 3b). While *Gli1* levels returned to baseline at day 7 after a single dose of SAG at 0 dpi, Hh activation was still significantly elevated when SAG was administered on day 4 with both injury models. Interestingly, we still detected high *Gli1* expression levels 5 days after SAG was administered 2 days post GLY injury, while in contrast, Hh activity was no longer elevated post CTX injury (Fig. 3b). These data suggest that the pharmacokinetic and/or -dynamic parameter of SAG-induced Hh activity differs between the two injury types.

We next evaluated the effects of sustained versus temporal Hh activation on IMAT formation through quantification of PERILIPIN$^+$ adipocytes. Confirming our and others previous work[45–47], GLY, compared to CTX, induced ~4-fold more IMAT (Fig. 3c). In the context of a CTX injury, sustained and temporal Hh activation significantly blocked IMAT formation, albeit to a lesser extent than when activated at 4 dpi (Fig. 3c). In contrast, Hh activation displayed a narrower therapeutic time window in the GLY model. For example, IMAT post GLY injury was inhibited the strongest with sustained Hh activation, followed by SAG administration at 2 dpi, with only a modest anti-adipogenic impact when injected at day 4 (Fig. 3c). In contrast to CTX, there was no effect on IMAT formation when SAG was administered at the time of GLY injury (Fig. 3c). These data indicate that the time window where FAPs are susceptible to Hh-induced adipogenic repression is broadest for a CTX injury with a very narrow therapeutic window between 2- and 4-days post injury for GLY.

To determine if Hh activity is also required for myofiber regeneration, we assessed the size of PHALLOIDIN$^+$ myofibers for various SAG dosing time points 7 days post CTX and GLY injury (Fig. 3d). In both injury models, sustained Hh activation and activation at day 4 post injury led to a decrease in average myofiber size, while no effect was observed when SAG was administered on the day of injury (Fig. 3d and Supplementary Fig. 3b). Unexpectedly, Hh activation at 2 days post injury had opposing effects depending on the type of injury

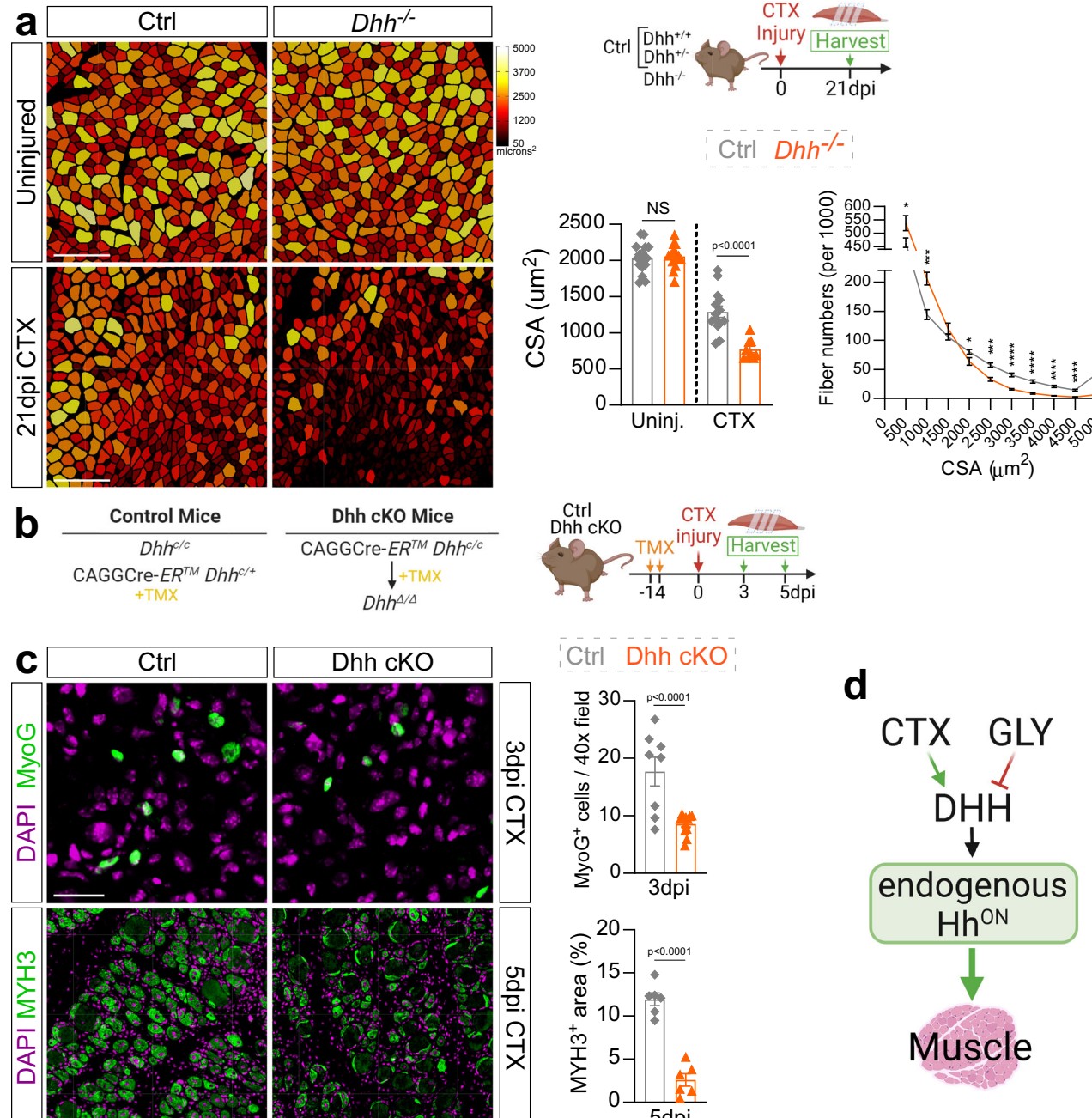

**Fig. 2 | Loss of *Dhh* impairs myofiber regeneration. a** Myofibers of uninjured and 21 days post CTX (cardiotoxin) injury from *Dhh⁻/⁻* and control (ctrl) mice are color-coded based on cross sectional area (CSA; μm²). Scale bar: 250 μm. *Bottom Right*: Average CSA (μm²) of uninjured (*n* = 16 TAs) and injured (*n* = 9 TAs) *Dhh⁻/⁻* compared to uninjured (*n* = 17 TAs) and injured (*n* = 14 TAs) control mice. Fiber number distribution based on CSA of *Dhh⁻/⁻* (*n* = 9 TAs) and ctrl mice (*n* = 14 TAs) 21 dpi post CTX. \**p* = 0.03, \*\*\**p* = 0.001 and \*\*\*\**p* ≤ 0.0001. Source data are provided as a Source

Data file. **b** Experimental outline. **c** Immunofluorescence and quantifications of MYOG⁺ myoblasts (green) in *Dhh⁻/⁻* (*n* = 11 TAs) and ctrl mice (*n* = 8 TAs) at 3 dpi (days post injury) post CTX. Scale bar: 25 μm. MYH⁺ myofibers (green) in *Dhh⁻/⁻* (*n* = 6 TAs) and ctrl mice (*n* = 6 TAs) at 5 dpi post CTX. Scale bar: 250 μm **d** *Model*: DHH is required for successful myofiber regeneration. Nuclei were visualized with DAPI (magenta). All data are represented as mean ± SEM. An unpaired two-tailed t test or a one-way ANOVA followed by a Dunnet's multiple comparison was used.

(Fig. 3d). Similar to two recent reports[54,55], SAG administration post CTX promoted regenerative myogenesis (Fig. 3d) indicating that boosting Hh levels during acute injuries is beneficial. However, SAG administration at 2 days post GLY injury severely impaired myofiber regeneration (Fig. 3d). Thus, ectopic Hh activation can act as a regenerative or degenerative signal depending on the injury model and time window (Fig. 3e).

Our data indicate that a single SAG injection at 2 dpi causes prolonged Hh activation post GLY injury compared to CTX (Fig. 3b). To determine if the defects of muscle regeneration we observed in the

GLY model are due to sustained, high-level Hh activity, we administrated SAG at 0-, 2- and 4 days post GLY injury at different doses: 2.5 mg/kg (1x dose) was further diluted to 1.25 mg/kg (0.5x dose); 0.83 mg/kg (0.3x dose); or 0.5 mg/kg (0.2x dose) (Supplementary Fig. 3c). Using RT-qPCR, we found a dose-dependent induction of *Gli1* expression (Supplementary Fig. 3c). Quantifying the number of PERILIPIN⁺ adipocytes, we detected a strong correlation between Hh levels and IMAT formation (Supplementary Fig. 3c). Similarly, myofiber size, based on CSA measurements, was also inhibited in a dose-dependent manner (Supplementary Fig. 3c). Thus, ectopic Hh

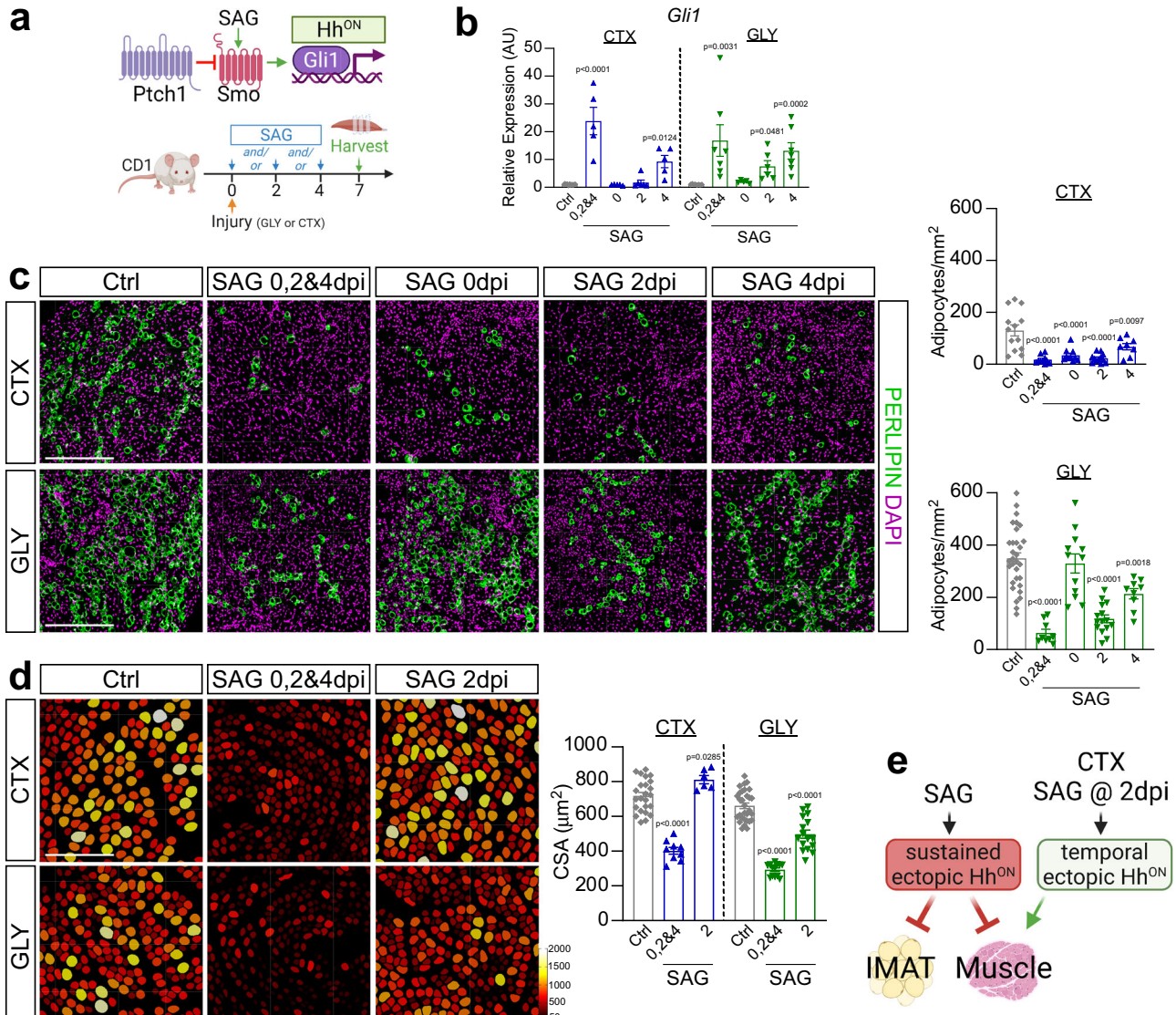

**Fig. 3 | Hh controls IMAT formation and myofiber regeneration in an injury- and temporal-dependent manner. a** Experimental outline. **b** RT-qPCR for *Gli1* 7 days post CTX (cardiotoxin) or GLY (glycerol) injury when treated with vehicle (CTX: $n = 11$; GLY: $n = 8$), SAG (Smoothened Agonist) at 0-, 2- and 4 days post injury (dpi) (CTX: $n = 5$; GLY: $n = 7$), at 0 dpi (CTX & GLY: $n = 5$), at 2 dpi (CTX & GLY: $n = 6$) or SAG at 4 dpi (CTX: $n = 5$; GLY: $n = 7$). **c** *Left:* Immunofluorescence for PERILIPIN⁺ adipocytes (green) 7 days post CTX *(top)* or GLY *(bottom)* injury when treated with vehicle, SAG at 0-, 2- and 4 dpi, SAG at 0 dpi, SAG at 2 dpi or 4 dpi. Nuclei were visualized with DAPI (magenta). Scale bars 250 μm. *Right:* Quantification of adipocytes per injured area (mm²) from CTX *(top)* and GLY *(bottom)* 7 days post injury. Vehicle control (CTX: $n = 13$; GLY: $n = 31$), SAG at 0-, 2- and 4 dpi (CTX: $n = 10$; GLY:

$n = 9$), SAG at 0 dpi (CTX & GLY: $n = 12$), SAG at 2 dpi (CTX: $n = 12$; GLY: $n = 15$) or SAG at 4 dpi (CTX: $n = 8$; GLY: $n = 9$). **d** *Left:* Color-coded myofibers based on cross sectional area (CSA) of CTX *(top)* and GLY *(bottom)* injured mice treated with SAG at 0-, 2- and 4 dpi, and SAG at 2 dpi. Scale bars: 250 μm. *Right:* Average CSA (μm²) at 7 dpi. Vehicle control (CTX: $n = 27$; GLY: $n = 30$), SAG at 0-, 2- and 4 dpi (CTX: $n = 9$; GLY: $n = 13$), SAG at 2 dpi (CTX: $n = 6$; GLY: $n = 17$). **e** *Model:* SAG-induced ectopic Hh activation blocks IMAT formation in both injuries. Sustained ectopic Hh activation also impairs muscle regeneration, while a single bolus of SAG at day 2 post CTX, but not GLY, improves regeneration. Each graph data point in b-d represents one TA muscle. All data are represented as mean ± SEM. One-way ANOVA followed by a Dunnet's multiple comparison was used.

activation post GLY injury blocks IMAT but also inhibits myofiber regeneration in a dose-dependent fashion.

## FAPs are the main cell type in skeletal muscle that detect and respond to DHH

As Hh impacts both adipogenesis and myogenesis, we next wanted to identify the cell type through which Hh executes both functions. Asking which cell types may be capable of inducing Hh signaling, we first mined the same scRNAseq data set as described above for cells expressing the Gli transcription factors *Gli1, Gli2* and *Gli3*. We found that FAPs, and to a lesser degree MuSCs, are the two main cell population that express all three Glis in muscle (Fig. 4a), and, therefore, would be able to execute Hh signaling. Fittingly, we and others have

previously reported that FAPs and MuSCs possess a primary cilium[8,55,60-63], a prerequisite to respond to Hh[30].

We next isolated FAPs and MuSCs using magnetic activated cell sorting (MACS) from *Dhh⁻/⁻* and SAG-treated mice 3 days post CTX injury and compared the induction of the Hh targets *Gli1* and *Ptch1* as a readout for Hh activity via RT-qPCR (Fig. 4b). As expected, we found that Hh activity is turned off in FAPs upon loss of *Dhh* and turned on upon SAG treatment. Surprisingly, we only detected weak Hh induction in MuSCs upon SAG treatment, while the expression of both Hh targets was unchanged upon loss of *Dhh* (Fig. 4b).

To determine whether the weak Hh activation in MuSCs has any functional impact on myofiber regeneration, we performed an in vitro myogenesis assay in the presence or absence of SAG using

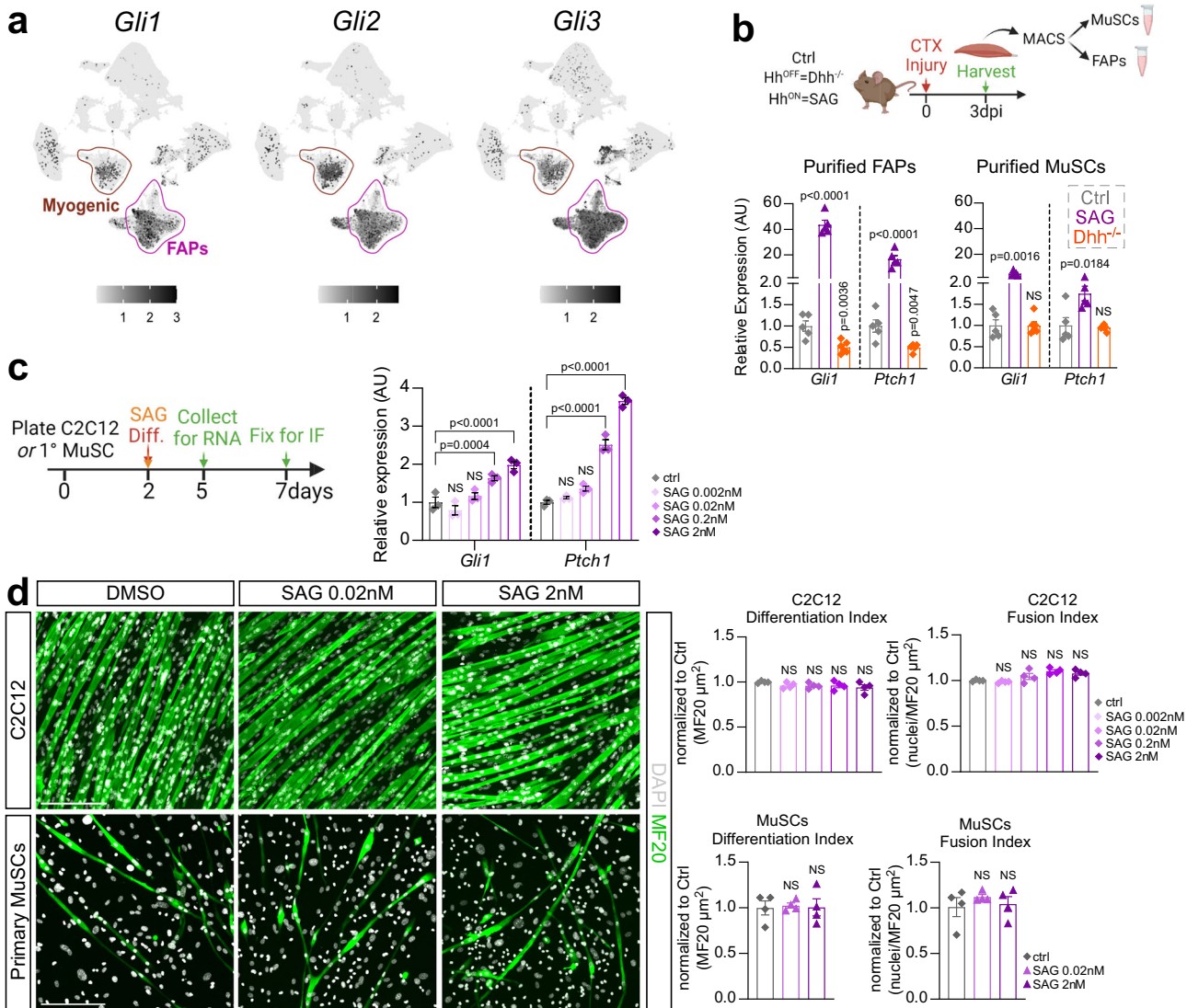

**Fig. 4 | FAPs are the main Hh responder. a** UMAP (Uniform Manifold Approximation and Projection) plots displaying aggregated log-normalized expression of the Hh transcription factors Gli1, Gli2 & Gli3. **b** RT-qPCR of expression of *Gli1* and *Ptch1* at 3 dpi CTX (cardiotoxin) from MACS- (magnetic activated cell sorting) isolated purified FAPs (*Left*) and MuSCs (*Right*) of control (ctrl) ($n = 5$), $Dhh^{-/-}$ ($n = 5$) and SAG- (Smoothened Agonist) treated mice ($n = 5$). **c** *Left*: Experimental design. *Right*: RT-qPCR of expression of *Gli1* and *Ptch1* of C2C12 ($n = 3$ replicates per experimental group) cells 5 days after induction of differentiation in ctrl and varying concentrations of SAG-treated cells (0.002 nM, 0.02 nM, 0.2 nM and 2 nM).

**d** *Left*: Immunofluorescence of mature myofibers labeled with MF20 (Myosin heavy chain; green) and nuclei with DAPI (white) 7 days post myogenic induction of C2C12 (*top*) or primary MuSCs (*bottom*). Cells were treated with either vehicle (DMSO) or SAG (0.02 nM and 2 nM). Scale bars: 200 μm. *Right*: Quantifications ($n = 4$ replicates for all experimental groups) of the differentiation index (MF20$^+$/μm$^2$) and fusion index (nuclei per MF20$^+$/μm$^2$) of C2C12 (*Top*) or primary MuSC-derived (*Bottom*) myofibers. All data are represented as mean ± SEM. One- way ANOVA followed by a Dunnet's multiple comparison was used.

C2C12 and primary MuSCs (Fig. 4c). Based on RT-qPCR for Hh target genes *Gli1* and *Ptch1*, SAG induced Hh signaling in C2C12 cells at increasing concentrations. However, similar to our in vivo results, the overall Hh response was weak. Next, we determined the impact of Hh activity on myofiber differentiation. Myofiber formation was evaluated through visualizing MF20$^+$ fibers (differentiation index), while rate of myoblast fusion was calculated by assessing number of myonuclei (DAPI) within MF20$^+$ fibers (fusion index). We found that SAG treatment did not affect differentiation or fusion of either C2C12 cells or primary MuSCs (Fig. 4d). Thus, our data demonstrate that FAPs are the main cellular Hh responder. At the same time, while MuSCs can respond to an ectopic Hh stimulus, albeit weakly, this has no impact on their differentiation into myofibers. Interestingly, our data also highlights that MuSCs are insensitive to the endogenous Hh ligand DHH.

## Sustained Hedgehog activation impacts FAP differentiation and survival

Based on our results so far, Hh must impact myogenesis indirectly through FAPs, the main cellular responder of endogenous Hh activity (Fig. 4). To genetically test this hypothesis, we used a conditional mouse model, which, upon Tamoxifen (TMX) administration, results in the genetic deletion of Ptch1, a negative regulator of the pathway, within FAPs (*Pdgfrα-Cre*ERT *Ptch1*$^{c/c}$, called FAP$^{no\ PTCH1}$). As we previously reported[8], loss of *Ptch1* results in FAP-specific ectopic Hh activation. Tamoxifen was administered to 12-week-old mice for two consecutive days by oral gavage, followed by a 2-week wash-out period before injuring TAs with GLY (Fig. 5a). We first evaluated IMAT formation by quantifying the number of PERILIPIN-expressing adipocytes (Fig. 5b). While there was no difference in the amount of IMAT in the absence of injury (Supplementary Fig. 4a), IMAT formation is robustly

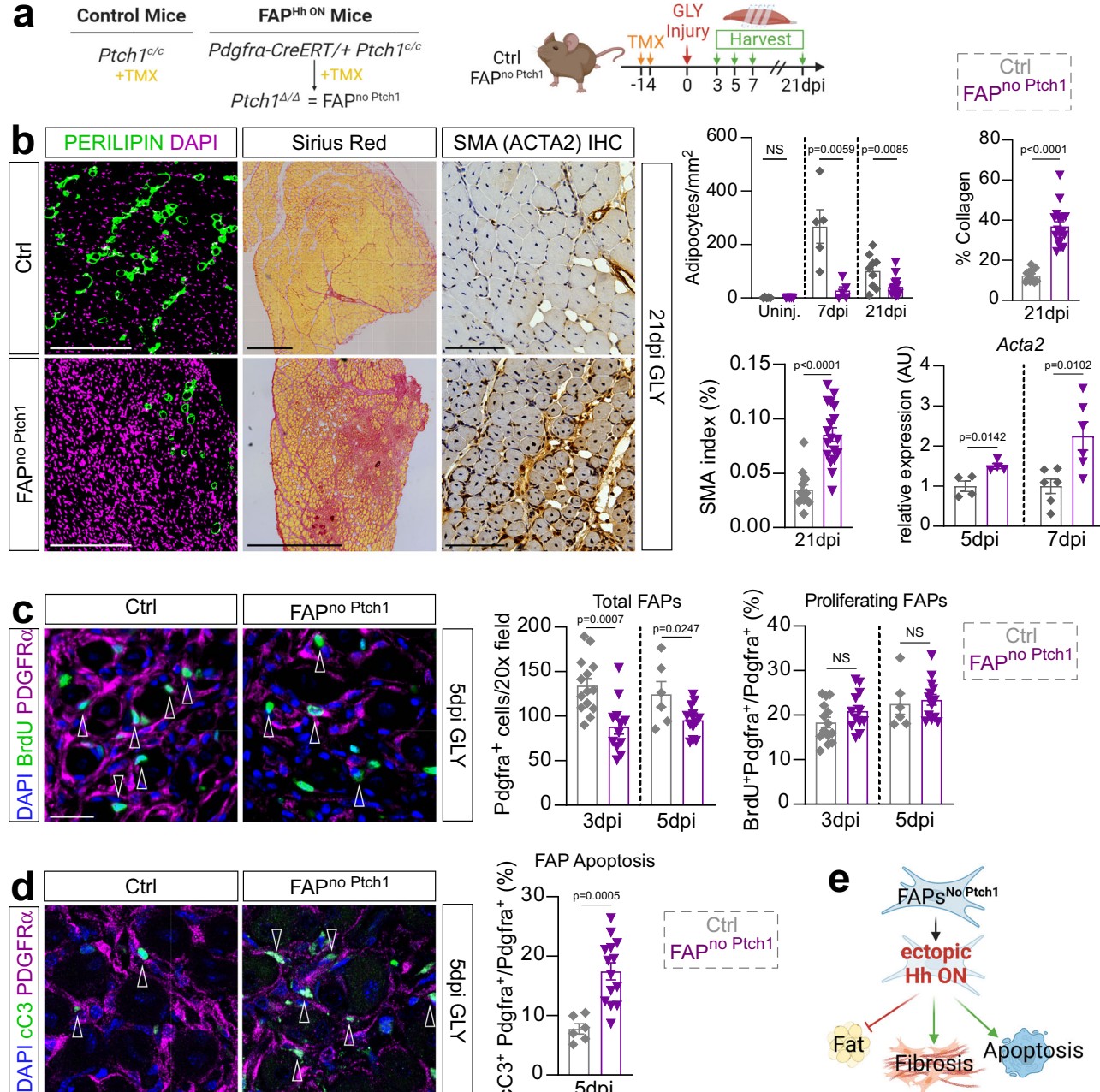

**Fig. 5 | Sustained high-level Hh activation blocks IMAT and promotes fibrosis.**
**a** Experimental outline. **b** *Left*: Immunofluorescence for PERILIPIN+ adipocytes (green) 21 days post GLY of ctrl and FAP^no Ptch1 mice. DAPI (purple) marks nuclei. Scale bars: 250 μm. *Middle:* Histological Sirius red staining. Scale bars: 500 μm. *Right:* Immunohistochemistry for αSMA (smooth muscle actin) at 21 dpi (days post injury) glycerol (GLY). Scale bars: 100 μm. *Far Right*: Quantification of adipocytes of uninjured (FAP^no Ptch1 n = 5; ctrl n = 5) mice, 7 days (FAP^no Ptch1 n = 5; ctrl n = 5) and 21 days post GLY (FAP^no Ptch1 n = 14; ctrls n = 9). Quantification of percent of Sirius red-positive collagen (red) of total TA area of FAP^no Ptch1 (n = 18) and control (n = 12) mice 21 dpi. Quantification of percent of αSMA+ cells (brown) of total TA area of FAP^no Ptch1 (n = 18) and control (n = 14) mice 21 dpi. RT-qPCR of expression of *Acta2* at 5dpi (n = 4 for both) and 7dpi (n = 6 for both). **c** *Left*: Immunofluorescence of BrdU+ cells (green) and FAPs (PDGFRα+ cells; magenta) 5 days post CTX (cardiotoxin)

injury in ctrl and FAP^no Ptch1 mice. Arrowheads point to double positive cells. DAPI (purple) marks nuclei. Scale bar: 25 μm. *Right*: Quantifications of total FAPs per 20x field at 3dpi (FAP^no Ptch1 n = 12; ctrl n = 14) and 5dpi (FAP^no Ptch1 n = 14; ctrl n = 6); and percent of proliferating FAPs (BrdU+ Pdgfra+ cells; % of total FAPs) at 3dpi (FAP^no Ptch1 n = 12; ctrl n = 14) and 5dpi (FAP^no Ptch1 n = 14; ctrl n = 6). **d** *Left*: Immunofluorescence of FAPs (PDGFRα+; magenta) undergoing apoptosis, marked by cleaved Caspase 3 (cC3+; green; arrowheads) 5 days post GLY in ctrl and FAP^no Ptch1 mice. DAPI (purple) marks nuclei. Scale bar: 25 μm. *Right*: Quantifications of percent of FAPs that undergo apoptosis (% of total FAPs) at 5dpi in FAP^no Ptch1 (n = 14) and ctrl (n = 6) mice. **e** *Model:* Ectopic activation represses the adipogenic conversion of FAPs, impacts their survival and promotes fibrosis. Each graph data point in (**b**–**d**) represents one TA muscle. All data are represented as mean ± SEM. An unpaired two-tailed t test or a one-way ANOVA followed by a Dunnet's multiple comparison was used.

blocked post GLY (Fig. 5b) and CTX injury (Supplementary Fig. 4b) in FAP^no PTCH1 mice compared to littermate controls. These results confirm and extend our previous findings that IMAT formation remains repressed beyond 7 days independent of injury type upon FAP-specific Hh activation[8].

In addition to adipocytes, FAPs can differentiate into myofibroblasts, the cellular origin of fibrotic scar tissue[7,15]. Since the fate of FAPs that have not differentiated into adipocytes is unknown, we asked if ectopic Hh activation may have pushed FAPs towards a fibrotic fate. For this, we evaluated fibrosis 21 days after a GLY and

CTX injury through a Sirius red staining and found a significant increase in collagen deposition in FAP$^{no\ Ptch1}$ compared to controls (Fig. 5b and Supplementary Fig. 4b). Next, we evaluated if the fibrosis is due to the differentiation of FAPs towards myofibroblasts. Fittingly, immunohistochemical (IHC) staining against α-Smooth muscle actin (Acta2, also called α-SMA), a marker of myofibroblasts, 21 days after a GLY injury showed a dramatic increase in α-SMA expressing cells in the FAP$^{no\ Ptch1}$ mice compared to controls (Fig. 5b). Confirming these findings, we found by RT-qPCR that there was already a significant increase in the expression of *Acta2* in FAP$^{no\ Ptch1}$ mice at earlier time points (Fig. 5b). These findings indicate that ectopic Hh activation may push FAPs to adopt a myofibroblast fate.

To define if this fibrotic conversion is dependent on the timing of Hh activation, we also quantified collagen deposition after temporal vs. sustained Hh activation through SAG post CTX and GLY injury (Supplementary Fig. 4c). Interestingly, we found increased Sirius red$^+$ collagen deposition 7 days post injury after sustained Hh activation (SAG given at 0-, 2- and 4dpi), while temporal activation (SAG given only at 2dpi) had no effect, independent of injury model (Supplementary Fig. 4c). This indicates that only sustained but not temporal Hh activation causes fibrosis.

While endogenous Hh activity had no impact on the proliferation or total number of FAPs (Fig. 1e), we wanted to elucidate whether ectopic Hh may be influencing FAP proliferation and/or survivability. Interestingly, we found that there is a decrease in the total number of PDGFRα$^+$ cells in FAP$^{no\ Ptch1}$ mice compared to controls 3- and 5dpi post GLY injury (Fig. 5c). However, we detected no differences in the proliferation rate of FAPs (BrdU$^+$ PDGFRα$^+$ cells) between genotypes (Fig. 5c). To determine whether increased cell death could explain this decrease in the overall number of FAPs, we co-stained FAPs for the apoptosis marker cleaved Caspase3 (cCas3). Interestingly, we saw a significant increase of cCas3$^+$ PDGFR-α$^+$ FAPs in FAP$^{no\ Ptch1}$ mice compared to controls at 5dpi (Fig. 5d) indicating that increased cell death is the most likely explanation for the reduced number of FAPs. Thus, sustained Hh activation causes increased cell death resulting in fewer FAPs. We propose that the remaining FAPs are then forced to adopt a myofibroblast instead of an adipogenic fate, causing massive scar tissue formation (Fig. 5e).

### FAP-specific Hh activation severely impacts regenerative myogenesis

While analyzing the FAP$^{no\ PTCH1}$ mice, we noticed that TAs from FAP$^{no\ Ptch1}$ mice appeared smaller in size compared to controls (Fig. 5b). Upon closer examination, while uninjured H&E-stained TAs were of comparable size between genotypes (Supplementary Fig. 5a), there was a significant decrease in the total area 21 days post GLY in TAs from FAP$^{no\ PTCH1}$ mice (Fig. 6a). To determine if the change in TA size is due to the loss of myofibers, we quantified the number of myofibers present in TA cross sections in FAP$^{no\ PTCH1}$ and control mice as a proxy for the total number of fibers present per TA (Fig. 6a). We failed to detect any differences in the number of muscle fibers indicating that myofiber regeneration is still functional in FAP$^{no\ PTCH1}$ mice (Fig. 6a). Next, we asked if the TA size differences are due to delayed or incomplete myofiber regeneration resulting in smaller myofibers by measuring individual myofiber CSAs between genotypes (Fig. 6a). There was no difference in CSA without injury (Fig. 6a and Supplementary Fig. 5a), highlighting that FAP-specific Hh activation is not required for myofiber homeostasis. In contrast, while control mice display significant myofiber recovery from 7 to 21 days post GLY injury, FAP$^{no\ PTCH1}$ mice failed to increase the size of their myofibers (Fig. 6a) and displayed a dramatic myofiber size shift towards smaller fibers (Supplementary Fig. 5b). This was also true 21 days post CTX injury (Supplementary Fig. 4c) indicating that the impact on myogenesis is independent of injury. We further quantified the myonuclear content of each regenerated myofiber and found that FAP$^{no\ Ptch1}$ mice

had significantly fewer myonuclei per fiber than controls (Fig. 6a). This suggests that ectopic FAP-specific Hh activation impairs myofiber regeneration in a cell non-autonomous fashion.

Upon muscle injury, MuSCs (PAX7$^+$ cells) become activated, expand, and transition along a stepwise process, controlled by pro-myogenic transcription factors such as MyoD and MyoG, to generate myoblasts. These myoblasts, in turn, continue to proliferate before differentiating into myocytes, and then fuse to either repair damaged or replace lost myofibers[3,64]. Therefore, we treated FAP$^{no\ PTCH1}$ and control mice with BrdU and assessed any differences in early myogenesis. Interestingly, we found that activation of Hh within FAPs led to a decrease in total number of PAX7$^+$ MuSCs due to defective proliferation of MuSCs as evident by the reduced frequency of BrdU$^+$ PAX7$^+$ cells compared to controls (Fig. 6b). Ectopic FAP-specific Hh activation also blocked the proliferation and expansion of MyoD$^+$ myoblasts (Fig. 6c) resulting in a smaller MyoG$^+$ myoblast pool (Fig. 6d). Collectively, our data demonstrate that ectopic, high-level Hh activation, indirectly through FAPs, suppresses the expansion of MuSCs resulting in a reduced myoblast pool and ultimately leading to smaller myofibers (Fig. 6e).

### DHH controls adipogenesis through TIMP3, while affecting myogenesis indirectly through FAP-produced GDF10

FAPs are crucial during muscle repair by secreting numerous beneficial factors[5]. Our data establish that FAPs are the main Hh responder and that the impact of Hh on both adipogenesis and myogenesis is through FAPs. To determine the molecular mechanism downstream of FAPs, we performed an enzyme-linked immunosorbent assay (ELISA) screen for multiple cytokines of whole muscle protein lysates from control and FAP$^{no\ Ptch1}$ mice (Supplementary Fig. 6a). We found no differences in >30 targets between genotypes, indicating that there are no gross alterations within the immune response when Hh is activated in a FAP-specific manner. Next, we evaluated the expression levels of known FAP targets (*Ccl2, Ccl7, Cxcl5, Fst, Gdf10, Igf1, Igf2, Il10, Il4, Il6, Spp1, Timp3* and *Wisp1*) that have been described to influence muscle regeneration and IMAT deposition in both FAP$^{no\ Ptch1}$ and *Dhh*$^{-/-}$ mice (Supplementary Fig. 6b). We identified 2 genes (Growth differentiation factor 10; *Gdf10* and Tissue inhibitor of metalloproteinase 3; *Timp3*) that were both significantly upregulated in FAP$^{no\ Ptch1}$ and down-regulated in *Dhh*$^{-/-}$ mice (Supplementary Fig. 6c). We further confirmed that FAPs are specifically and differentially expressing *Gdf10* and *Timp3* downstream of the Hh pathway by MACS-isolating PDGFRα$^+$ FAPs 3 days post CTX injury from control, SAG treated and *Dhh*$^{-/-}$ mice (Fig. 7a). Compared to control mice, in the absence of DHH, *Gdf10* and *Timp3* are significantly downregulated within FAPs (Fig. 7a). In contrast, upon ectopic Hh activation via SAG, FAPs significantly upregulate both *Gdf10* and *Timp3* (Fig. 7a). Thus, *Gdf10* and *Timp3* are expressed by FAPs, and their expression is dependent on Hh activity levels.

We previously identified TIMP3 as a FAP-specific Hh-dependent anti-adipogenic factor[8]. In our present study, we demonstrate that the endogenous expression of *Timp3* is also dependent on Hh activity. To functionally test whether loss of TIMP3 activity in *Dhh*$^{-/-}$ mice is the cause for the increased IMAT formation, we designed a rescue experiment where we restored TIMP3 function via the small molecule Batimastat in *Dhh*$^{-/-}$ mice. Batimastat is a pan metalloproteinase inhibitor, which, as we have previously shown, can act as a pharmacological TIMP3 mimetic and potently inhibits IMAT formation[8]. Batimastat was administered to control and *Dhh*$^{-/-}$ littermates, and at 7 days post CTX injury IMAT formation was evaluated (Fig. 7b). As expected, we saw an increase in PERIPIN$^+$ adipocytes and a decrease in myofiber CSA in *Dhh*$^{-/-}$ compared to control mice in the vehicle treated group (Fig. 7c). Excitingly, Batimastat treatment was able to block PERILIPIN$^+$ adipocyte infiltration in *Dhh*$^{-/-}$ mice. Batimastat has also been shown to improve muscle function in *mdx* mice, a mouse model of Duchenne Muscular Dystrophy[65]. Therefore, we also assessed any improvement

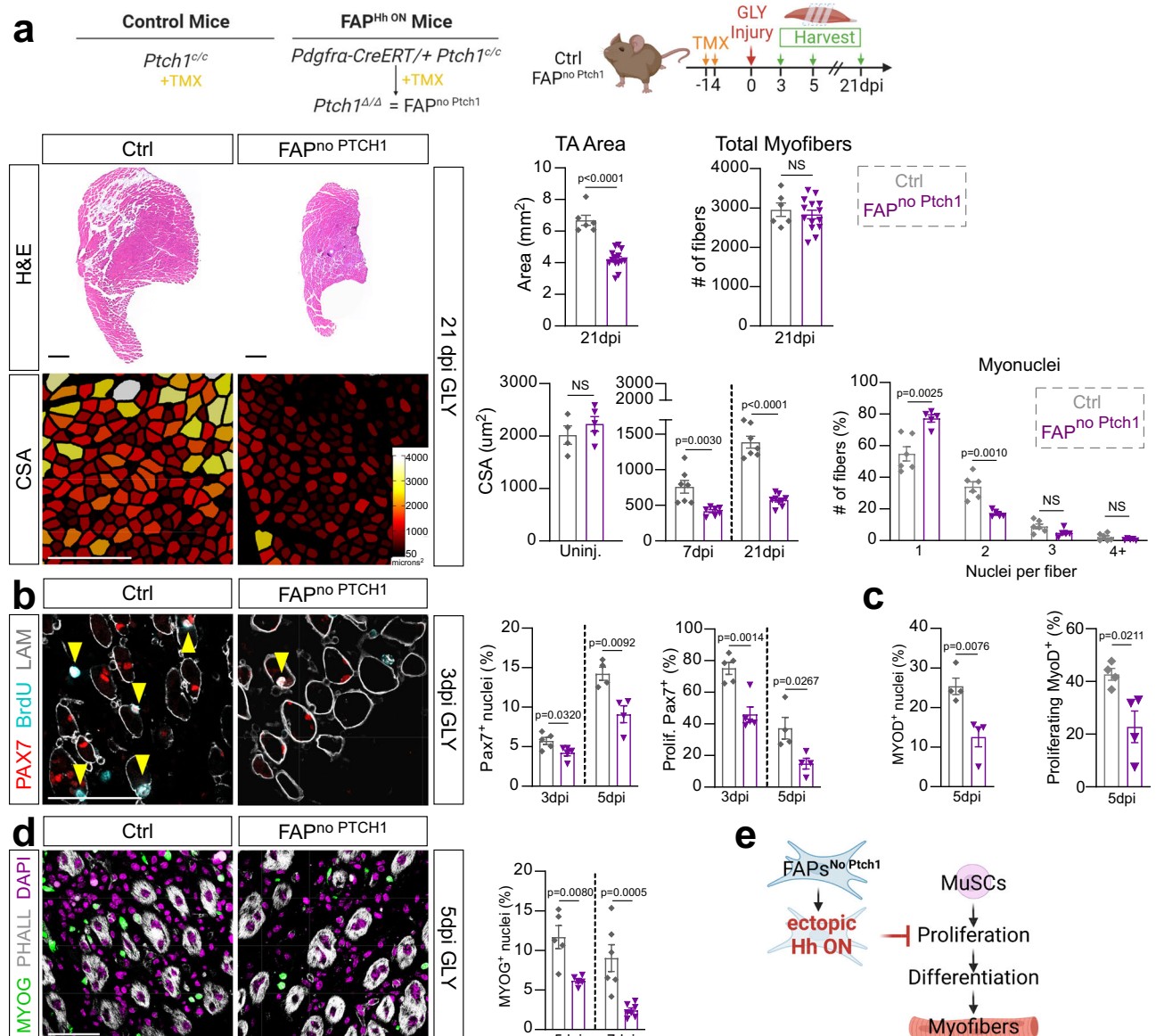

**Fig. 6 | FAP-specific Hh activation inhibits MuSC expansion and proliferation.** **a** *Left*: Hematoxylin and Eosin (H&E) staining of FAP[no Ptch1] and ctrl mice 21 dpi (days post injury). Scale bars: 500 μm. Myofibers color-coded according to size of FAP[no Ptch1] and ctrl mice 21 dpi. Scale bars: 250 μm. *Right*: Quantification of total TA area 21 days post GLY (glycerol) of FAP[no Ptch1] ($n = 14$) and ctrl ($n = 6$) mice. Quantification of total fiber numbers 21 dpi of FAP[no Ptch1] ($n = 14$) and ctrl ($n = 6$) mice. Average CSA (cross sectional area) of uninjured (FAP[no Ptch1] $n = 5$; ctrl $n = 4$), 7 dpi (FAP[noPtch1] $n = 7$; ctrl $n = 7$) and 21 dpi (FAP[no Ptch1] $n = 9$; ctrl $n = 7$) mice. Distribution of number of myonuclei per fiber (% of total fibers) from FAP[no Ptch1] ($n = 5$) and ctrl ($n = 6$) mice 21 dpi. **b** *Left*: Immunofluorescence for PAX7[+] (MuSCs (muscle stem cells), red), BrdU[+] (proliferating cells, cyan) and LAMININ (myofiber outline, white) 3 dpi. Double positive cells marked by arrowhead. Scale bar: 50 μm. *Right*: Quantifications of percent of PAX7[+] nuclei (% of total nuclei) and proliferating PAX7[+] MuSCs (% of total MuSCs) at 3 dpi ($n = 5$ for each) and 5 dpi ($n = 4$ for each). **c** Percent of MYOD[+] and BrdU[+] MYOD[+] myoblasts at 5 dpi in FAP[no Ptch1] ($n = 4$) and ctrl ($n = 4$) mice. **d** *Left*: Immunofluorescence of MYOG[+] nuclei (green) and PHAL-LOIDIN (white) 5 dpi. Nuclei visualized by DAPI (magenta). Scale bar: 50 μm. *Right*: Percent of MYOG[+] nuclei at 5 dpi ($n = 5$ for each) and 7 dpi (FAP[no Ptch1] $n = 7$ TAs; ctrl $n = 6$ TAs). **e** *Model*: Ectopic Hh activation in FAPs indirectly represses proliferation and expansion of MuSCs resulting in a reduced myoblast pool and smaller myofibers. Each graph data point in a-d represents one TA muscle. All data are represented as mean ± SEM. An unpaired two-tailed t test or a one-way ANOVA followed by a Dunnet's multiple comparison was used.

---

of myofiber regeneration in Batimastat-treated *Dhh*[−/−] mice. However, we detected no improvement in myofiber CSA compared to Batimastat-treated controls (Fig. 7c). Thus, DHH controls IMAT formation, but not myofiber regeneration, through TIMP3.

GDF10 is a member of the transforming growth factor-β (TGFβ) superfamily and has been recently identified as a pro-myogenic factor[10] secreted by FAPs[10,66]. Therefore, we next determined if FAP-derived GDF10 controls myogenesis. We first confirmed long-term transcriptional activation by GDF10 by measuring the expression of *Serpine 1* (also known as Pai1), a known transcriptional target of

GDF10[67], 3 days post treatment of C2C12 cells with recombinant GDF10 (rGDF10) (Supplementary Fig. 6d). Next, we treated isolated primary MuSCs with rGDF10 during myogenic differentiation (Fig. 7d). After 7 days in culture, we calculated myofiber differentiation and fusion indices through visualization of mature myofibers (MF20[+] fibers) along with nuclei (DAPI) (Fig. 7d). We found that there was an increase in both differentiation and fusion of myoblasts when primary MuSCs were treated with rGDF10 protein (Fig. 7d). Together, our results indicate that endogenous Hh activation, via its ligand DHH and sensed

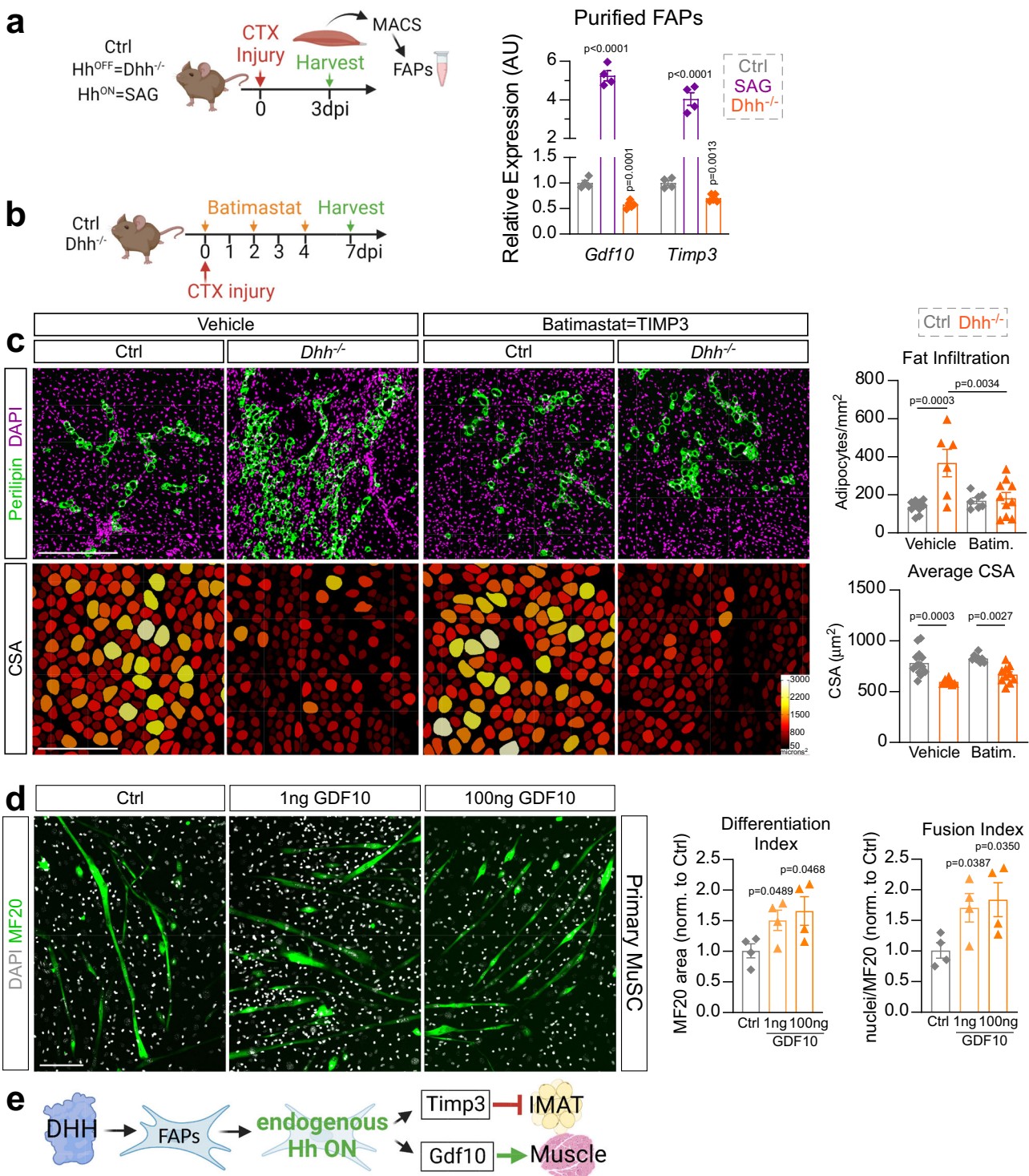

**Fig. 7 | DHH controls IMAT through TIMP3, and myogenesis, indirectly, through GDF10. a** RT-qPCR of *Gdf10* and *Timp3* within MACS- (magnetic isolated cell sorting) isolated FAPs (3dpi (days post injury) CTX (cardiotoxin)) from ctrl (*n* = 4 mice), *Dhh*⁻/⁻ (*n* = 6 mice) and SAG- (Smoothened agonist) treated mice (at 0- and 2dpi, *n* = 4 mice). **b** Experimental design. **c** *Left*: Immunofluorescence for PERILIPIN⁺ adipocytes (green) and DAPI⁺ nuclei (magenta). PHALLOIDIN⁺ myofibers (color coded based on cross sectional area (CSA)) 7 days post CTX injury of vehicle or batimastat treated *Dhh*⁻/⁻ and ctrl mice. Scale bars: 250 µm. *Right*: Quantification of number of adipocytes per injured area of vehicle-treated *Dhh*⁻/⁻ (*n* = 6) and ctrl (*n* = 10) in comparison to batimastat-treated *Dhh*⁻/⁻ (*n* = 10) and ctrl (*n* = 7) mice. Average CSA (µm²) of myofibers 7 dpi of vehicle-treated *Dhh*⁻/⁻ (*n* = 7) and ctrl (*n* = 14) in comparison to batimastat-treated *Dhh*⁻/⁻ (*n* = 10) and ctrl (*n* = 8) mice. Each graph data point represents one TA muscle. **d** *Left*: Immunofluorescence of MF20⁺ (myosin heavy chain) myofibers (green) differentiated from primary MuSCs (muscle stem cells), 7 days post induction and treated with GDF10 (1 ng and 100 ng) or control vehicle (PBS). Nuclei were visualized with DAPI (white). Scale bar: 150 µm. *Right*: Quantifications (*n* = 4 replicates for all experimental groups) for the differentiation (MF20⁺/µm²) and fusion index (nuclei per MF20⁺/µm²). **e** *Model*: After an acute injury, DHH activates Hh in FAPs, inducing expression of Timp3 and Gdf10, which in turn inhibit adipogenesis or promote myogenesis, respectively. All data are represented as mean ± SEM. A one-way ANOVA followed by Dunnet's multiple comparison was used.

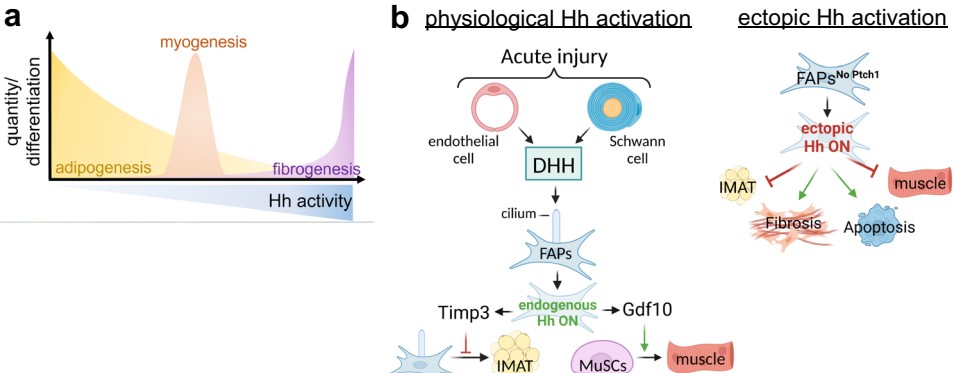

**Fig. 8 | Endogenous Hh signaling, via its ligand DHH, is crucial for balancing muscle regeneration and fatty fibrosis. a** *Hh activity levels control multiple cellular events*: Hh is a potent endogenous anti-adipogenic brake and its activity levels negatively correlate with IMAT formation. In addition, too little (loss of physiological Hh) or too much (ectopic) Hh activity impairs muscle regeneration, while only a short pulse of ectopic Hh activity at the right time boosts regeneration. In addition, sustained high-level Hh activation causes extensive fibrosis. **b** *Cellular model*: Following an acute but not adipogenic injury, endothelial and Schwann cells induce physiological Hh activation through the Hh ligand DHH, which is then sensed by cilia on FAPs. Endogenous Hh activation within FAPs leads to expression of Timp3, which blocks adipogenesis cell autonomously. Additionally, FAP-specific Hh activation controls myogenesis through Gdf10-dependent cell non-autonomous mechanism. In contrast, ectopic and sustained Hh activation within FAPs, while potently blocking their differentiation into IMAT, also causes depletion of the FAP pool, massive fibrosis and severely blocks myofiber regeneration.

by FAPs, balances adipogenesis through TIMP3 and myogenesis through GDF10 (Fig. 7e).

## Discussion

Increased IMAT infiltration with age and disease is strongly associated with decreased muscle strength and function[18,20–29]. However, it is still unclear whether IMAT forms due to the progressive loss of an adipogenic brake or the gradual gain of a pro-adipogenic trigger. We and others have previously demonstrated that ectopic Hh activation is sufficient to block adipogenesis[8,54]. Here, we define the physiological role of the Hh pathway, and identify Hh as an endogenous pathway that, through its ligand DHH, is necessary to restrict IMAT formation and promotes muscle regeneration post injury (Fig. 8). This significant discovery indicates that the pathological increase in IMAT may be due to loss of this adipogenic brake. At the same time, we show that the activity levels of Hh have a dramatic impact on the fate and function of FAPs. For example, while loss of endogenous Hh signaling promotes the adipogenic conversion of FAPs, ectopic high-level activation blocks adipogenesis. In addition, ectopic Hh activation results in fibrosis possibly by pushing FAPs to adopt a myofibroblast fate. We also demonstrate that Hh signaling both promotes (physiological Hh activation) and restricts (ectopic Hh activation) muscle regeneration indirectly through FAPs, and that the myogenic control of Hh is dependent on the level and timing of activation, as well as the type of injury. Together, we describe innovative cell autonomous and non-autonomous roles for Hh signaling in controlling adipogenesis, fibrogenesis and post-regenerative myogenesis. This has major implications for the development of therapeutics for tissues affected by fatty fibrosis.

The Hh ligands, SHH and DHH, have both been proposed to be involved in skeletal muscle[35–38,40]. However, which Hh ligand is responsible for muscle regeneration remained unclear. By querying scRNAseq data sets, we identified DHH as the main ligand being expressed by endothelial and Schwann cells independently and objectively confirming our and others previous work[8,38,42,43]. Recently, DHH has been shown to be required for neo-angiogenesis post ischemic hindlimb injury[38,43]. However, it was unknown if DHH is also necessary for IMAT formation and/or muscle regeneration after an acute muscle injury. Using a genetic DHH null and conditional mouse model and two independent Hh antagonists, we found that DHH is the main ligand that induces Hh signaling in skeletal muscle. Once active,

DHH restricts IMAT formation and promotes myofiber regeneration. Supporting our data, Kang et al. recently described that the cytokine Interleukin 15 (IL15) acts upstream of DHH and induces its expression[68]. Fittingly, IL15 activation also represses IMAT and enhances myofiber regeneration phenocopying active Hh signaling. Together, our results demonstrate that DHH acts as an endogenous adipogenic brake and fulfills a potent pro-myogenic role, albeit indirectly through FAPs, during postnatal tissue repair.

How does DHH balances adipogenesis vs. myogenesis? The two main cell populations that carry a primary cilium, and thereby would be able to respond to DHH, are FAPs and MuSCs[8,55,60,62,63]. Interestingly, our data demonstrate that FAPs are the main cellular responder of endogenous and ectopic Hh signaling. Once active, Hh induces the potent anti-adipogenic and secreted factor TIMP3 in FAPs, thereby blocking their adipogenic differentiation. In contrast, loss of Hh activity results in repression of TIMP3 and enhanced adipogenesis. Fitting with our results, recent data demonstrated that the pharmacological TIMP3 mimetic Batimastat can be successfully used to repress IMAT formation in various other conditions such as hindlimb ischemia[69] and limb girdle muscular dystrophy 2B[18] pointing to TIMP3 as a potential anti-IMAT therapy. In contrast, ectopic Hh signaling only induces a weak transcriptional response in MuSCs, while MuSCs are refractory to endogenous Hh activity. Recent data suggest that MuSCs use their cilia in vivo to maintain their quiescence[55,62,70]. However, the data are conflicting on whether MuSCs cilia sense active Hh signaling[55,70] or are required to keep the Hh pathway turned off[62]. While more experiments are needed to fully unravel the role of Hh signaling in MuSCs, our data provide compelling evidence that Hh controls muscle regeneration cell non-autonomously via its main cellular responder, FAPs. Supporting evidence was provided from a recent cell ablation approach[54]. Ablating Hh-responsive FAPs from muscle resulted in impaired muscle regeneration indicating that Hh does induce pro-myogenic factors in FAPs. Screening for FAP-specific myogenic factors that could be controlled by Hh, we successfully identified GDF10, which acts as a potent pro-myogenic factor. Interestingly, GDF10 expression declines with age and restoring its levels prevented the sarcopenic muscle phenotype[10] underscoring the importance of GDF10 for muscle health.

At the same time, the FAP[no PTCH1] data also conflict with our previous results where we found that FAP-specific, but low-level, Hh activation enhances muscle regeneration[8]. There, we demonstrated

that cilia guard GLI3 repressor levels in FAPs. More specifically, removing FAP cilia resulted in loss of GLI3-R leading to Hh de-repression and low-level Hh activation[8]. Similar to our FAP[no PTCH1] data, this resulted in de-repression of TIMP3, which in turn inhibited the adipogenic differentiation of FAPs via changes to the ECM composition[8]. However, low-level Hh activation improved muscle regeneration[8]. This is in direct contrast to high-level Hh activation in our FAP[no PTCH1] mice, which, compared to our previous studies, resulted in severely compromised myogenesis. To determine if the levels of Hh activation matter, we attempted to mimic low versus high Hh activity levels by administering our Hh agonist at various concentrations. However, we failed to detect any beneficial myogenic effects at lower doses. Therefore, it is likely that Hh de-repression versus full Hh activation may induce different myogenic factors that either positively or negatively affect myogenesis. Fittingly, the number of genes being repressed in the Hh OFF state is much larger than the ones that are being induced in the Hh ON state[71]. It will be interesting to fully define the FAP secretome in the Hh ON, OFF and de-repressed states.

We also found that ectopic Hh activation, either global or specifically in FAPs, dramatically increased the number of myofibroblasts resulting in massive scar tissue formation. In contrast, we did not observe any changes in fibrosis when we turned Hh off by removing *Dhh*. Thus, while Hh is sufficient to induce fibrosis, it is not necessary. As fibrotic scar tissue is a potent anti-myogenic signal[11,72,73], this increased fibrosis could also contribute to the myogenic defects we observe after ectopic Hh activation. Together, chronic high-level Hh activation induces muscle fibrosis. However, the role of Hh in tissue fibrosis needs further clarification as Hh signaling can act both as an anti-fibrotic and a pro-fibrotic signal depending on the tissue and type of injury[39,74–77]. For example, genetic lineage tracing of FAPs from control and FAP[no PTCH1] mice would determine if FAPs are shifted away from an adipogenic to a fibrogenic fate, thereby becoming myofibroblasts.

When is DHH required during the regenerative process? By administrating the Hh agonist SAG at different time points post injury, we were able to determine the time windows for when Hh is required. We found that, although CTX has an earlier Hh-sensitive window, Hh activation at 2 days post GLY and CTX injury caused the largest repression of IMAT. Interestingly, our data also demonstrated that activation of Hh at day 2 affected fibrosis the most in both injuries suggesting that the influence of Hh on the fate of FAPs is strongest at day 2. This timing fits with our previous observations that early pro-adipogenic genes can be detected as early as 3 dpi with the first visible lipid droplets by day 5[8,78]. This suggests that ectopically elevating Hh signaling pushes FAPs from an adipogenic to a fibrogenic fate resulting in reduced IMAT and increased fibrosis. We also find that Hh activation has both positive and negative effects on muscle regeneration depending on injury type and time. For example, SAG administration at days 2 and 4 post GLY injury impairs muscle regeneration. In contrast, Hh activation at day 2 post CTX boosted myofiber regeneration, similar to findings from a recent report[54], while SAG at day 4 after CTX inhibited myogenesis similar to GLY. One possible explanation for this is that, because Hh signaling is already being induced after a CTX injury, the system is primed to respond to Hh compared to GLY, where Hh is being inactivated ([8] and Supplementary Fig. S2c). Boosting Hh activity via SAG may therefore enhance its beneficial effects. In addition, CTX and GLY cause different kinetic responses, which could explain the differences in timing requirement and phenotypic consequences. For example, CTX causes an earlier immune[79] and pro-fibrotic[45] response compared to GLY. Similarly, while CTX displays stronger myofiber degeneration at early injury stages, myofiber regeneration is less efficient with GLY[45,46]. Together, our data indicates that Hh signaling has a very narrow potential therapeutic window that is dependent on injury type.

This work reveals that the Hh pathway, through its ligand DHH, is sufficient and necessary to restrict IMAT formation during muscle regeneration. The gradual loss of this endogenous adipogenic brake provides an attractive explanation for why pathological IMAT forms especially as Hh activity is severely blunted with age and disease[35–37,39,55]. In addition, the extent of Hh activity might represent a potent predictor on how much IMAT is allowed to form. We also uncovered that Hh acts, depending on its activity levels, both as a pro- and anti-myogenic signal and controls muscle regeneration through both cell autonomous and non-autonomous mechanisms. Thus, the Hh pathway remains an attractive therapeutic avenue, but further studies are needed to determine optimal dosing and timing.

# Methods

## Animal studies

All mouse alleles used in this study have been reported before. B6;129S-*Dhh*[tm1Amc]/J (also called Dhh[−/−] mice, Jax# 002784) mice were kindly provided by Humphrey Yao and maintained on a 129S1/SvlmJ background. As *Dhh*[+/+] and *Dhh*[+/−] mice displayed no phenotypic differences (Fig. S1), we mostly used *Dhh*[+/−] mice as controls allowing us to optimize our breeding strategy. As the fat phenotype was more prominent in females and we failed to detect any sex differences (Fig. S1d), we mostly focused on female mice. For conditional deletion of DHH, we used the global deleter allele CAGGCre-ERTM (Jax# 004682) crossed to a floxed Dhh allele (*Dhh*[lox/lox]; kindley provided by Drs. Ange-Marie Renault and Russell Norris and previously described[43]). For genetic manipulation and targeting of the Hh pathway, we used *Ptch1*, a negative inhibitor of the Hh pathway, specifically from FAPs using a tamoxifen-inducible, *Pdgfra*[CreERT2] allele[80] (Jax# 032770) crossed to *Ptch1*[tm1Bjw] allele[81] (Jax# 030494). PDGFRA is the "gold-standard" marker in the field to identify FAPs within murine and human muscle[7–9,12–15,18,82]. To induce deletion of target genes, Tamoxifen (TRC, T006000) was dissolved in corn oil and administered through oral gavage (200–250 mg/kg) on two consecutive days. We waited two weeks post TAM administration before performing injuries. All mice independent of genotype were gavaged with tamoxifen. Littermates lacking the *Cre* allele or being heterozygous for the *floxed* allele were used as controls. For all SAG experiments, CD1 or 129S1/SvlmJ wildtype mice were used, either kindly provided by Margaret Hull (UF) or purchased through Charles River. To maintain the same genetic background as the *Dhh*[−/−] mice, for Hh loss of function experiments utilizing Gant61 and Vismodegib, 129S1/SvlmJ mice were used. Mice were housed in standard ventilated cages at controlled temperature (22–23 °C), 40–50% humidity, 12-h light/dark cycle, and ad libitum access to food and water. All animal work was approved by the Institutional Animal Care and Use Committee (IACUC) of the University of Florida.

## Injuries and small molecules

Muscle injuries to the Tibialis Anterior (TA) were induced with 30–50 μL of either 50% glycerol (GLY, Acros Organics, 56-81-5) or 30–50 μL of 10 μM cardiotoxin (CTX; *Naja Pallidum*, Latoxan, L8102-1MG) diluted in sterile saline. All injuries were performed on 10–14 week-old adult mice (additional details specific to each experiment are listed in *Results* section). After injury, various small molecules were administered to mice to manipulate the Hh pathway (Vismodegib, Gant61 and SAG) or to mimic TIMP3 function (Batimastat). Vismodegib (Apex Bio; Cat. No: A3021) was formulated as a suspension in 0.5% methylcellulose and 0.2% Tween-80. Upon injury and for the next 4 consecutive days, Vismodegib was administered at 166 mg/kg concentration through oral gavage. Gant61 (MedChemExpress, Cat. No: HY-13901) was resuspended in EtOH to create a stock solution. Upon the day of injury (0 dpi) and at 2 dpi, 150 mg/kg solution diluted in sterile saline was administered subcutaneously. Batimastat (Apex Bio, cat. No: A2577) was resuspended in DMSO to make a stock solution.

The stock was further diluted to obtain a 3 mg/kg solution in sterile saline and 0.01% Tween-80; administered through intraperitoneal (IP) injection on the day of injury, 2- and 4 dpi. For SAG 21k treatment (Tocris, Cat. No: 5282) a stock solution was made in DMSO. This stock was further diluted in sterile saline and administered at 2.5 mg/kg through IP injection on day 0-, 2- and 4 dpi; or alternatively, a single dose on either the day of injury, or 2 dpi, or 4 dpi. For the dose experiments, the 2.5 mg/kg (1x dose) was further diluted to 1.25 mg/kg (0.5x dose); 0.83 mg/kg (0.3x dose); or 0.5 mg/kg (0.2x dose). All control mice received the corresponding vehicle solution lacking the drug.

### Histology, immunohistochemistry and image analysis

Immunostaining and imaging was done as previously outlined[78]. In brief, upon harvesting, TAs were cut in two parts: the proximal third was used for RNA isolation, while the distal two thirds of the TA was used for histology and immunostaining. TAs were fixed in 4% PFA (Paraformaldehyde) for 2.5 h at 4 °C, washed and cryoprotected overnight in 30% sucrose. TAs were placed in OCT-filled cryomolds (Sakura; 4566) and frozen in LN2-cooled isopentane. 3-4x cryosections per TA were collected with a Leica cryostat at 10–12 μm thickness every 250–350 μm. Sections were incubated with primary antibodies in blocking solution (5% donkey serum in PBS with 0.3% Triton X-100) overnight at 4 °C. Primary antibodies used were rabbit anti-Perilipin (1:1000; Cell Signaling, 9349 S), rabbit anti-Laminin (1:1000; Sigma-Aldrich, L9393), rabbit anti-MyoG (1:250; Proteintech Group 14688-1-AP), rabbit anti-cleaved Caspse 3 (1:500, Millipore Sigma AB3623), and goat anti-PDGFRα (1:250, R&D Systems #AF1062). Antigen retrieval using a Sodium Citrate Buffer (10 mM Sodium Citrate, 0.05% Tween-20, pH 6.0) was required for mouse anti-PAX7 (1:15; DSHB, AB 428528, supernatant), rat anti-BrdU (1:1000; Abcam AB6326), rabbit anti-MYH3 (1:250, Proteintech 22287-1-AP) and rat anti-MyoD (1:250; Invitrogen, MA1-41017). Alexa Fluor-conjugated secondary antibodies from Life Technologies (1:1000) in combination with the directly conjugated dyes Phalloidin-Alexa 568 and 647 (1:200, Molecular Probes # A12380 & A22287) were added for 1 h at room temperature and slides were mounted for imaging (SouthernBiotech; 0100-01). DAPI (Invitrogen, D1306) was used to visualize nuclei. Staining of fibrillar collagen was done through a Sirius red staining[83], and areas occupied by Sirius Red were quantified through ImageJ Software, using the Color-Threshold function. Immunohistochemical (IHC) staining of α-SMA⁺ cells (1:500, Sigma A2547) was done following the manufacturer's instructions (Vector Laboratories, DAB Substrate Kit, SK-4100 and Vectastain Elite ABC Kit, PK-6100). After dehydration, slides were counter-stained with Hematoxylin (1:10) and mounted with Permount (Fisher Chemical, SP15-100). Areas occupied by α-SMA⁺ cells were quantified through ImageJ Software, using the Color-Threshold function. Hematoxylin and Eosin (H&E) staining was done to visualize the TA area. Images from randomly chosen fields across several sections (each 250-350 μm apart) were acquired with a Leica DMi8 microscope equipped with a SPE confocal and a high-resolution color camera. The navigator function within the Leica LSA software was used to generate cross-sectional images of whole tibialis anterior muscles. All images were processed identically and quantified with Fiji/ImageJ (v1.52p). To quantify total and proliferating MuSC (PAX7 & BrdU) and differentiating MuSCs (MYOD & MYOG), 4–6 areas were randomly selected using a 40x objective and imaged between 3–5 μm of Z resolution. Total proliferating FAPs (PDGFRα & BrdU) was calculated by imaging 3–5 areas with a ×20 objective, and total number of cells within that area were averaged. We calculated the average CSA using our recently published image segmentation pipeline[46]. Briefly, sections were stained against PHALLOIDIN or LAMININ to visualize myofibers. TA images were processed through ImageJ Software to exclude uninjured areas, identified as areas where myofibers lack centralized located nuclei. These images were first processed through Cellpose and then through our

LabelsToRois ImageJ plug-in[46]. IMAT was calculated by quantifying individual adipocytes, visualized through PERILIPIN staining, and normalized to injured TA area ($mm^2$) as previously described[78]. For in vitro myofibers, cells were fixed with 4% PFA for 15 min at room temperature (RT), washed and incubated with blocking solution for 1 h. Primary antibodies used were MF20-s (1:20; DSHB, 2147781). For each well, 5 images (2mmx2mm) were taken and the average of them were plotted. Fiji/ImageJ software was used to quantify the differentiation and fusion index. The differentiation index was obtained by thresholding the MF20 channel, followed by measuring the total area through the Analyze particle function. The fusion index was calculated by additionally thresholding total DAPI, applying the Watershed function to separate individual nuclei, quantifying the number of nuclei through the Analyze particle function. Thereafter, Image Calculator was used to identify the total number of nuclei within fibers and quantified through the Analyze particle function.

### Single-cell RNA sequencing analysis

Publicly available single-cell RNA sequencing data were downloaded and prepared as in McKellar et al. 2021[41]. All code used to prepare and analyze these data is available on github (https://github.com/mckellardw/scMuscle) and fully preprocessed data are available for download on Dryad (doi:10.5061/dryad.t4b8gtj34). Briefly, 111 single-cell and single-nucleus RNA sequencing datasets were downloaded, aligned to the mm10 genome using cellranger (v3.1.0), preprocessed with Seurat (v3.2.1)[84], and integrated using Harmony (v1.0)[85]. Cells were clustered and annotated according to expression of 112 canonical marker genes, which can be found alongside the code used in these analyses. Cells were then subset to only include injury timepoints for which at least 5000 cells/nuclei were available. These timepoints included uninjured, 0.5, 1, 2, 3.5, 7, 10, and 21 days post-injury. Datasets were then mined for expression of the three Hh ligands and the three GLI transcription factors and results displayed as aggregated data from all datasets.

### Expression analysis

Muscle tissue was homogenized in TRIzol (ThermoFisher Scientific, 15596026) using a bead beater (TissueLyser LT, Qiagen, 69980) at 50 Hz for 5 min, with one 5 mm (Qiagen, 69989) and three 2.8 mm metal beads (Precellys, KT03961-1-101.BK). Following the TRIzol instruction manual, chloroform was added to isolate the upper, RNA-containing phase[78]. RNA was then purified utilizing the QIAGEN RNeasy kit per manufacturer's instructions. 500–800 ng of RNA was transcribed into cDNA using the qScript Reverse Transcriptase kit (Quanta bio; 84003). RT-qPCR was performed in technical quadruplets on a QuantStudio 6 Flex Real-Time 384-well PCR System (Applied Biosystems, 4485694) using PowerUp SYBR Green Master Mix (ThermoFisher Scientific, A25742). Fold changes were calculated using the $2^{-\Delta\Delta CT}$ method[86] and expression levels were normalized to the housekeeping genes *Hprt, Sra1* and *Pde12*. Primer sequences are provided in Supplementary Table 1.

### Mouse cytokine assay

TAs were lysed using a TissueLyser LT (Qiagen, 85600) with 5 mm (Qiagen, 69989) and 2.8 mm (Precellys, KT03961-1-101.BK) stainless steel beads in 1x PBS only including proteinase inhibitor (Pierce, Cat #A32953). After centrifugation, the supernatant was collected, and the protein concentration was determined using a BCA assay (Pierce, Cat #23225). Samples were then run for an ELISA assay (32-plex mouse cytokine assay, Eve technology).

### In vitro Assays

For in vitro myogenesis, C2C12 cells (ATCC CRL-1772 and[87]) and primary MuSCs were cultured at 37 °C in 5% CO2 in DMEM (Gibco 11965-092) containing 10% FBS (Invitrogen 10438026) and 1% GlutaMax

(Gibco 35050-061). On day 2, myogenesis was induced by switching from FBS to 2% horse serum (Gibco 26050-070) with media change every two days. For expression analysis, RNA was collected 3 days after differentiation, while immunofluorescence studies were done 7 days after differentiation. Primary MuSCs isolation was done as previously described[88]. In brief, TAs and Gastrocnemius muscles from 129S1/SvlmJ mice were injured with CTX (50- and 80 μL, respectively) and harvested 3 days post injury. Muscles were mechanically and enzymatically dissociated to a uniform slurry in digest media containing DMEM (Gibco, 10566024), 1% Pen/Strep, 50 mg/mL collagenase IV (Worthington, LS004188), and 6 U/mL Dispase (Gibco, 17105041) for 1 h at 37 °C with constant agitation. Primary MuSCs were then plated in gelatin-coated 24 well plates and differentiated into myotubes with 2% horse serum as described above. To induce Hh signaling during myogenesis, SAG (Tocris, Cat. No: 5282) or the vehicle DMSO was added to C2C12 and primary MuSCs starting at day of induction at varying concentrations (0.002 nM, 0.02 nM, 0.2 nM or 2 nM) as indicated. Full length mouse GDF10 protein (Sino Biological, 50165-M01H) or vehicle (sterile PBS) was added to primary MuSCs at start of myogenic induction at 1 ng or 100 ng.

## Magnetic activated cell sorting

Magnetic activated cell sorting (MACS) was performed according to manufacturer instruction and as previously described[8]. Briefly, injured Tibialis anterior muscles were harvested at 3 days post injury and mechanically and enzymatically dissociated with Collagenase II (Gibco 17101-015; 0.15% final) and Dispase II (Sigma D4693-16, 0.04% final) in DMEM. After filtration (70 um filter; Fisher 22363548), we first isolated FAPs using the CD140a (PDGFRα) MicroBead Kit (Miltenyi Biotec 130-101-502). The flow-through was then subjected to MuSC enrichment using the satellite cell isolation kit (Miltenyi Biotec 130-104-268). Once each population was isolated, cells were resuspended in RLT buffer followed by RNA isolation (Qiagen RNeasy Micro Kit 74004) and RT-qPCR analysis as outlined above.

## Statistical analysis

TAs with injury less than 50% of total area were excluded from this study. The experimenter was blinded until data were collected. All data was graphed using GraphPad Prism (version 9) with data presented as mean ± SEM. For comparing two samples with one variable, an unpaired two-tailed $t$ test was used. For more than two samples with one variable, a one-way ANOVA followed by a Dunnett's multiple comparison test was used. For two variables, a two-way ANOVA followed by Tukey's multiple comparison was carried out. A $p$ value less than 0.05 was considered statistically significant and denoted as follows: *<0.05, **<0.01, ***<0.001 and ****<0.0001.

## Reporting summary

Further information on research design is available in the Nature Portfolio Reporting Summary linked to this article.

## Data availability

Source data are provided with this paper. The code used to derive all scRNAseq data is available on github (https://github.com/mckellardw/scMuscle) and fully preprocessed data are available on Dryad (https://doi.org/10.5061/dryad.t4b8gtj34). All other correspondence and material requests should be addressed to dkopinke@ufl.edu. Source data are provided with this paper.

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

## Acknowledgements

The authors thank the members of the Kopinke laboratory for helping with data collection and critical reading of the manuscript. We also thank Karyn Esser for her valuable input on the manuscript. We greatly appreciate the sharing of CD1 mice by Margaret Hull, the *Pdgfra^CreERT* allele by Gabrielle Kardon and the Dhh flox mice by Russell Norris. This study would have been impossible without Colin Dinsmore, who was the first to ask, "So, which Hh ligand is responsible?". Last, we are immensely grateful for the unwavering support and mentoring of Jeremy Reiter. This work was supported by the US National Institutes of Health (NIH) grants NIH grant 1R01AR079449 to DK, R01AG058630 to BDC, T32HD043730 to AMN and T32EB023860 to DWM. The content is solely the responsibility of the authors and does not necessarily represent the official views of the NIH. All schematic figure models were created with BioRender®.

## Author contributions

A.M.N. and D.K. designed the experiments and wrote the manuscript. A.M.N., A.B.A., C.D.J., and L.Y.Z performed the experiments and analyzed data. M.R. provided the Dhh lox mice. D.H. provided primary MuSCs and provided helpful comments. D.W.M. and B.D.C. designed and analyzed the RNAseq data. All authors discussed the results and commented on the manuscript.

## Competing interests

The authors declare no competing interests.
