## [Peer Review File · Nature Communications]

REVIEWER COMMENTS

Reviewer #1 (Remarks to the Author):

Norris et al. demonstrate that Hh activation is an important determinant of adipogenic fate of fibroadipogenic progenitors (FAP) in the muscle, and that dysregulation of Hh in FAPs can also alter muscle regeneration. The experiments are presented clearly, and this reviewer does not have any major issues with the claims supported by the data. However, the main question for my review is whether there are significant conceptual advances presented by this manuscript to be considered for publication Nature Communications. The findings that hedgehog is involved in adipogenesis in the muscle has been covered by the senior author's prior work. The potentially more interesting data presented here is that Hh alteration of adipogenesis also alters muscle regeneration (which is also kind of covered by prior work, but to a less degree), but there is little mechanistic insight presented here to really advance our understanding of how Hh is modifying muscle repair through the alteration of adipogenic fate. This is an important question for the authors to address as mechanistic insight on how Hh is directly or indirectly altering muscle repair through FAPs or muscle stem cells would represent conceptual advancement to merit stronger considerations for publication.

Major points

1. While Fig. 1 and 2 nicely demonstrates the source of hedgehog activation during muscle injury repair and the role of HH signaling in FAPs in determining adipogenic fate, many of these findings were similarly described in the previous paper from Kopinke et al., Cell, 2017. Some thematically redundant data includes the previous description that DHH is the main hedgehog ligand and produced in the Schwann cells (Fig. 4H,I of previous paper) and that hedgehog activation in FAPs (through ptch1 deletion and HH agonists) inhibits adipogenic fate (Fig. 4D,E of previous paper). It is not clear if Fig 1 and 2 in the new paper really moves the conceptual framework forward other than validating what was already shown using different genetic model and compounds.
2. Fig 3 nicely demonstrates that the loss of Hh activation in association with increased adipogenesis impairs muscle regeneration (at least in the CTX model, but not GLY injury model, with rationale presented that is not clear to this reviewer), but the mechanism for this is left as an open question by this manuscript, with Fig 4 providing essentially negative data showing that HH-induction of TIMP3 does not alter muscle regeneration. This is a really important question for this manuscript, due to the lack of conceptual novelty from the first two figures. Potential mechanism includes some other non-cell autonomous effect from the FAPs, or cell autonomous effect on muscle stem cells (which are also activated by Hh), none of which is really explored in this paper.
3. Fig 5 carefully maps on the temporal requirement of Hh agonists in modifying muscle regeneration, but without fundamental understanding of how Hh is modifying muscle regeneration, the findings of Fig 5 is diminished.

4. Fig 6 demonstrated that cell-autonomous HH activation of FAPs are also deleterious for muscle regeneration, accompanied by the loss of myogenic differentiation. As pointed out in the previous comment, the fact that Hh can alter muscle regeneration is a key finding of this manuscript, but there is little to no mechanistic insight from these studies other than the observation that alteration of adipogenesis is associated with modification of muscle regeneration, which is not entirely novel based on the author's prior work.

5. Fig 7 showed that cell-autonomous and pharmacologic HH activation of FAPs is also accompanied by tissue fibrosis. This suggests that a very tenuous balance of Hh is required to maintain muscle repair (too little = bad, too much = bad), and that targeting Hh itself is perhaps not the best pharmacologic strategy for muscle repair. What is confusing is the presentation of the data that cilia knockout does not produce the same phenotype in this figure, even though the authors previously reported that cilia knockout itself induces Hh activation in FAPs. The explanation that the difference is due to the degree of Hh activation induced in the different model system is not really backed up by experimental data.

Minor points

1. Title is a too vague for this research article, and dilutes any potential impact.
2. "Failed tissue regeneration often results in fatty fibrosis" in the abstract lacks rigorous clinical context. Outside of fatty liver disease (nonalcoholic steatohepatitis, or NASH), and maybe muscular dystrophy as demonstrated by the authors (although one could argue whether that is a true fibrotic disease), there are few precedents where this reviewer can identify strong association of adipose tissue with fibrotic parenchymal changes, unless the authors can provide more references to that effect.

Reviewer #2 (Remarks to the Author):

In this manuscript the authors exploited a variety of approaches to modulate Hedgehog (Hh) signaling pathway systemically or specifically in fibro/adipogenic progenitors (FAPs). The results show that DHH is the ligand that regulates Hh-mediated inhibition of intramuscular fat (IMAT) formation and increased muscle regeneration. Surprisingly, and contrary to authors previous data and expectations, the conditional activation of Hh in FAPs inhibits adipogenesis, but also promotes fibrotic scar tissue and inhibits muscle regeneration.

This reviewer appreciates the number of approaches used to investigate the role of Hh signaling in balancing tissue regeneration and fatty/fibrotic degeneration in injured muscles. The data are clear and fairly support the authors conclusions on difference in outcomes of muscle regeneration depending on magnitude and timing of Hh signaling activation. However, there are important open issues in term of

mechanism by which Hh exerts these effects that should be better addressed by the authors experimentally – see points below.

- 1) From author data it is hard to determine how systemic Dhh KO influences MuSC behavior - directly or through indirect modulation of other signals. The author can address, at least partly this important issue, by testing whether genetic or pharmacological inhibition of Hh in MuSCs influence the ex vivo ability to proliferate and differentiate of MuSCs isolated from the models of muscle injury by CTX or GLY. Alternatively, the authors can use MuSCs associated to single fibers from the same experimental models
- 2) Co-culture studies between FAPs no PTCH1 (and control wt) and MuSCs isolated from injured (or unperturbed) muscles and treated or not with TMX, in various combinations, are required to determine how Hh in FAPs regulates MuSC biology.
- 3) Also, mono-cultures of FAPs isolated from the experimental mouse models and conditions should be used to independently determine FAPs phenotype ex vivo after cultures

Reviewer #3 (Remarks to the Author):

In this paper the authors identify the hedgehog family member that is involved in muscle regeneration, confirm its effects on interstitial fat deposition and expand their analysis of this pathway to show a defect in myofiber regeneration. They further identify a downstream mediator on the effect on adipogenicity, and they show that it does not mediate the effect on fiber regeneration. This is an interesting and informative piece of work, and while much could be added to clarify a number of arising questions, it already contains enough findings to be a significant contribution to the field. I was particularly impressed by the fact that the authors managed to separate the adipogenic phenotype from the defect in myofiber regeneration, which is a classical chicken-and-egg problem in the field.

The results presented in figure one are apparently compelling, but more information should be provided: If the Umap plots presented are the results of aggregating a number of publicly available dataset, how many of the datasets are contributing to the results displayed? As an example, it is evident that Gli1,2 and 3 ae expresses in a small minority of the FAPs and myogenic cells in figure 1c. Is this minority originating from one of the aggregated datasets, or from multiple? In particular Gli3 in FAP{s appears to follow an alternating patterns, with higher frequency of expressing cells (albeit lower absolute levels of expression) on days 1, 3.5 and 10, while showing a different patterns (higher expression in rare cells) in the rest of the time course. Is it because the data from days are contributed by a specific set of sequencing samples? This needs to be clarified.

The results in figure 5, in which the authors explore the timing of activation of the pathway, are interesting but not too surprising, as it seemed unlikely that DHH would act after the initial wave of apoptosis that reduces FAPs numbers following acute damage. Still, it is interesting to observe that the positive effects of this pathway's activation on fibers is also taking place at about the same time, as this supports the proposal that the effect on myofibers is also mediated by FAPs. It would be great if the authors could include in the text a statement on the pharmacokinetics of the smoothed agonist in vivo, ideally in mice, as that would help interpreting the results.

The results presented in figure 6/7 are the most exciting of the manuscript, suggesting that DHH indeed works through FAPs. Indeed the defect in regeneration in animals in which FAPs lack ptch is impressive however in order to conclude that this is due to a differentiation block, the authors should really exclude an effect on survival or expansion. This is particularly important given that Pax7 cells did not change in numbers or BrdU positivity. BrdU should be shown for MyoD positive and ideally MyoG positive cells as well (less important as the drop in numbrs takes place between the Pax7 and myoD stages). Given the significant drop in MyoD positive cell numbers, and the lack of differences in the proliferation of Pax7 cells, what happened to the "missing" cells? A differentiation block without changes in proliferation or survival should lead to an accumulation of cells at a more immature stage.

I was also very surprised that the effect of knocking out ptch on FAP numbers, proliferation and survival was not addressed at all... it seems like an important parameter. Finally, it is not clear to me why the authors looked at the myogenic defect only with glycerol damage... is this effect also present when CTX is used?

Reviewer #1: Norris et al. demonstrate that Hh activation is an important determinant of adipogenic fate of fibroadipogenic progenitors (FAP) in the muscle, and that dysregulation of Hh in FAPs can also alter muscle regeneration. The experiments are presented clearly, and this reviewer does not have any major issues with the claims supported by the data. However, the main question for my review is whether there are significant conceptual advances presented by this manuscript to be considered for publication Nature Communications.

We were heartened to read that this reviewer “does not have any major issues with the claims supported by the data.” We added a substantial amount of new data that (a) establish FAPs as the main cellular responder of Dhh, (b) define the cellular phenotype of FAPs and MuSCs in response to Hh, (c) demonstrate that the function of Hh signaling on myogenesis is indirect through FAPs, and (d) identify Gdf10 as the myogenic factor produced by FAPs in response to Hh activation. In addition, we have extensively rewritten the manuscript to emphasize the multiple conceptual advances of our work.

The findings that hedgehog is involved in adipogenesis in the muscle has been covered by the senior author’s prior work.

Our previous work only used genetic and pharmacological gain-of-function models, and, therefore, only showed that Hh is sufficient to repress adipogenesis. In this work, we uncover that endogenous Hh signaling is also necessary to prevent the adipogenic conversion of FAPs to FAT, thus, acting as an endogenous adipogenic brake in muscle. This is highly innovative as it is still unclear how intramuscular fat forms. In addition, we provide the first functional data to demonstrate that DHH is the key Hh ligand in muscle that activates Hh signaling, settling a long-standing debate. To address this concern, we have highlighted the novelty of our work throughout the revised manuscript.

The potentially more interesting data presented here is that Hh alteration of adipogenesis also alters muscle regeneration (which is also kind of covered by prior work, but to a less degree), but there is little mechanistic insight presented here to really advance our understanding of how Hh is modifying muscle repair through the alteration of adipogenic fate. This is an important question for the authors to address as mechanistic insight on how Hh is directly or indirectly altering muscle repair through FAPs or muscle stem cells would represent conceptual advancement to merit stronger considerations for publication.

We agree with the reviewer and have spent considerable time and resources to address this interesting question. We now provide evidence that DHH is sensed mainly by FAPs and that MuSCs only display a minimal transcriptional response to Hh (Fig. 4b & c). Importantly, we provide direct evidence that myogenesis of either C2C12 or primary MuSCs is not impacted by Hh signaling (Fig. 4d). Our new Figure 7 highlights that FAPs produce, in response to Hh activation, Timp3 to control adipogenesis and Gdf10 to regulate myogenesis. We have also added more insights into how endogenous Hh regulates myogenesis

on the cellular level (Figs. 2c & 6b-d). Thus, Hh influences muscle regeneration indirectly through FAP-produced GDF10.

Major point 1: While Fig. 1 and 2 nicely demonstrates the source of hedgehog activation during muscle injury repair and the role of HH signaling in FAPs in determining adipogenic fate, many of these findings were similarly described in the previous paper from Kopinke et al., Cell, 2017. Some thematically redundant data includes the previous description that DHH is the main hedgehog ligand and produced in the Schwann cells (Fig. 4H,I of previous paper) and that hedgehog activation in FAPs (through *ptch1* deletion and HH agonists) inhibits adipogenic fate (Fig, 4D,E of previous paper). It is not clear if Fig 1 and 2 in the new paper really moves the conceptual framework forward other than validating what was already shown using different genetic model and compounds.

We thank the reviewer for this comment. In response, we have now combined parts of the original Figures 1 and 2 into a new Figure 1, added additional data explaining how loss of DHH affects FAPs, and rewritten the first paragraph to highlight our novel findings, creating an overall stronger rationale for why we functionally addressed the role of the Hh ligand DHH during muscle regeneration.

To address the first critique more specifically, the use of single cell RNAseq allowed us to objectively inquire about the temporal and spatial expression of all three Hh ligands. Our findings demonstrate that *Dhh* is indeed dynamically being induced upon injury, independently confirming our previous results obtained through RTqPCR of whole muscle lysate (Kopinke et al., Cell, 2017). However, they also highlight the various cell populations that express DHH, which is novel. For example, while this indeed includes Schwann cells as we have previously reported, our data also point to endothelial cells as an important cellular source.

Regarding the second comment, the original Figure 2 defined the function of DHH and necessity of endogenous Hh signaling during intramuscular fat formation, both completely novel concepts. Thus, while our previous data demonstrated that ectopic Hh activation is sufficient to repress adipogenesis (Kopinke et al., Cell, 2017), the data presented here determine that Hh is also necessary to repress fat formation and, thus, acts as an endogenous adipogenic brake during muscle regeneration. As the reasons for why intramuscular fat forms are still unclear, our data provide an exciting novel explanation.

2. Fig 3 nicely demonstrates that the loss of Hh activation in association with increased adipogenesis impairs muscle regeneration (at least in the CTX model, but not GLY injury model, with rationale presented that is not clear to this reviewer), but the mechanism for this is left as an open question by this manuscript, with Fig 4 providing essentially negative data showing that HH-induction of TIMP3 does not alter muscle regeneration. This is a really important question for this manuscript, due to the lack of conceptual novelty from the first two figures. Potential mechanism includes some other non-cell autonomous effect from the FAPs, or cell autonomous effect on muscle stem cells (which are also activated by Hh), none of which is really explored in this paper.

We thank the reviewer for this comment. To address this, and as outlined above, we now provide new data assessing the cellular impact of lack of Hh activity on MuSCs (Fig. 2c). Next, we established that FAPs, and not MuSCs, are the key cellular responder of DHH and that Hh activation has no impact on *in vitro* myogenesis of either C2C12s or primary MuSCs (Fig. 4b-d). We then moved all data from the original Figure 4 to the new Figure 7, and added new data from a screen for FAP-secreted factors that are being controlled by Hh signaling (Figs. 7 & S6). We also provide functional evidence that Hh regulates myogenesis through secretion of Gdf10 from FAPs (Fig. 7d). Thus, we now provide compelling evidence that Hh controls myogenesis through a non-cell autonomous effect.

The distinction between a CTX and a GLY injury model is an important one, and we apologize for not making this as clear as possible. We have included new data that demonstrate that a CTX injury induces Hh via activation of DHH, while a GLY injury represses Hh (Fig. S2c). Thus, loss of DHH does not further alter Hh activity levels post GLY injury, thereby not causing any additional increase in intramuscular fat.

This is an important control experiment that indicates that loss of Hh activity, the adipogenic brake, may be the cause for the massive amount of intramuscular fat that forms post GLY compared to CTX, where Hh is being induced.

3. Fig 5 carefully maps on the temporal requirement of Hh agonists in modifying muscle regeneration, but without fundamental understanding of how Hh is modifying muscle regeneration, the findings of Fig 5 is diminished.

To address this concern, we have generated multiple new lines of evidence resulting in the new Figure 4. Here, we demonstrate that, while MuSCs can weakly respond to the Hh agonist SAG, FAPs are the main responder (Fig. 4b & c). We further determined that this weak Hh response to SAG had no impact on myogenesis using both C2C12 and primary myoblasts (Fig. 4d). We also provide additional data on the kinetics of SAG demonstrating that Hh is being induced within 6-12hrs after SAG administration (Fig. S3a). Thus, we believe that, combined with the addition of the new data, our Hh agonist results provide meaningful insights into the (a) differential phenotypic responses to Hh activation, (b) temporal and dosage requirements of Hh, and (c) key cellular responders of Hh. Our findings are especially important, and broadly applicable, as such a rigorous analyses have been missing in prior studies that have used Hh agonists in various *in vivo* experiments.

4. Fig 6 demonstrated that cell-autonomous HH activation of FAPs are also deleterious for muscle regeneration, accompanied by the loss of myogenic differentiation. As pointed out in the previous comment, the fact that Hh can alter muscle regeneration is a key finding of this manuscript, but there is little to no mechanistic insight from these studies other than the observation that alteration of adipogenesis is associated with modification of muscle regeneration, which is not entirely novel based on the author's prior work.

As outlined above, we have spent a lot of effort into addressing this interesting question. For example, we now provide evidence that Hh controls myogenesis indirectly through FAPs and not MuSCs (Figs. 4 & 6). In addition, we added new data that establish the molecular mechanisms on how FAP-specific Hh activation controls adipogenesis (Timp3) vs. myogenesis (Gdf10) (Fig. 7).

5. Fig 7 showed that cell-autonomous and pharmacologic HH activation of FAPs is also accompanied by tissue fibrosis. This suggests that a very tenuous balance of Hh is required to maintain muscle repair (too little = bad, too much = bad), and that targeting Hh itself is perhaps not the best pharmacologic strategy for muscle repair. What is confusing is the presentation of the data that cilia knockout does not produce the same phenotype in this figure, even though the authors previously reported that cilia knockout itself induces Hh activation in FAPs. The explanation that the difference is due to the degree of Hh activation induced in the different model system is not really backed up by experimental data.

We fully agree with the reviewer's statement regarding the tenuous balance of Hh activity in skeletal muscle. We also acknowledge the fact that including the fibrosis data from our FAP-specific cilia knockout mice may have been confusing. To focus on, and strengthen, our argument that ectopic Hh activation prevents FAPs from adopting an adipogenic fate and instead pushes them towards a fibrogenic fate, we have streamlined our existing data and performed numerous new experiments. First, we removed the cilia knockout fibrosis data, as cilia are not the main focus of this paper. Next, we focused on assessing the fate of FAPs upon Hh activation. Our new data demonstrate that sustained Hh signaling induces cell death of FAPs. In addition, the remaining FAPs are then pushed towards a fibrogenic fate and differentiate into SMA+ myofibroblasts, the cellular origin of fibrosis (Fig. 5b-d). Thus, ectopic Hh activation, when sustained over time, represses intramuscular fat but induces fibrosis by causing FAPs to turn into myofibroblasts.

Minor points

1. Title is a too vague for this research article, and dilutes any potential impact.

We appreciate this feedback and have changed our title as follows: "Hedgehog signaling via its ligand DHH acts as cell fate determinant during skeletal muscle regeneration".

2. "Failed tissue regeneration often results in fatty fibrosis" in the abstract lacks rigorous clinical context. Outside of fatty liver disease (nonalcoholic steatohepatitis, or NASH), and maybe muscular dystrophy as demonstrated by the authors (although one could argue whether that is a true fibrotic disease), there are few precedents where this reviewer can identify strong association of adipose tissue with fibrotic parenchymal changes, unless the authors can provide more references to that effect.

We have removed this sentence and changed the abstract accordingly.

Reviewer #2: In this manuscript the authors exploited a variety of approaches to modulate Hedgehog (Hh) signaling pathway systemically or specifically in fibro/adipogenic progenitors (FAPs). The results show that DHH is the ligand that regulates Hh-mediated inhibition of intramuscular fat (IMAT) formation and increased muscle regeneration. Surprisingly, and contrary to authors previous data and expectations, the conditional activation of Hh in FAPs inhibits adipogenesis, but also promotes fibrotic scar tissue and inhibits muscle regeneration. This reviewer appreciates the number of approaches used to investigate the role of Hh signaling in balancing tissue regeneration and fatty/fibrotic degeneration in injured muscles. The data are clear and fairly support the authors conclusions on difference in outcomes of muscle regeneration depending on magnitude and timing of Hh signaling activation. However, there are important open issues in term of mechanism by which Hh exerts these effects that should be better addressed by the authors experimentally – see points below.

We greatly appreciate the time, kind words and valuable suggestions to improve our manuscript.

1) From author data it is hard to determine how systemic Dhh KO influences MuSC behavior - directly or through indirect modulation of other signals. The author can address, at least partly this important issue, by testing whether genetic or pharmacological inhibition of Hh in MuSCs influence the ex vivo ability to proliferate and differentiate of MuSCs isolated from the models of muscle injury by CTX or GLY. Alternatively, the authors can use MuSCs associated to single fibers from the same experimental models

As suggested by the reviewer, we now provide evidence that Hh signaling indirectly influences myogenesis through FAPs. For example, we have MACS-isolated FAPs and MuSCs from Dhh null and SAG treated animals and found that FAPs are the main responder of Hh (Fig. 4b). Additionally, and as suggested by the reviewer, we cultured C2C12 and primary MuSCs and found no difference in either the fusion or differentiation index upon Hh activation with SAG (Fig 4d). Thus, while MuSCs display weak Hh activation (Fig 4c), it does not impact their differentiation. We also provide new evidence that lack of DHH results in a drastically reduced pool of myoblasts leading to delayed myofiber regeneration (Fig. 2c). Thus, Hh controls myogenesis indirectly through FAPs. To note, we focused on ectopic Hh activation instead of inhibition as we have failed to detect endogenous Hh activation in either primary FAPs or MuSCs, most likely as Hh ligands are not present in the serum.

2) Co-culture studies between FAPs no PTCH1 (and control wt) and MuSCs isolated from injured (or unperturbed) muscles and treated or not with TMX, in various combinations, are required to determine how Hh in FAPs regulates MuSC biology.

To address how FAP-specific Hh activation could impact MuSC biology, we screened previously known FAP-specific factors that affect myogenesis in Dhh null, FAP-no-Ptch1 and SAG treated mice *in vivo*. We found that FAP-specific Hh activation induces Timp3 and Gdf10, whereas both factors are repressed when Hh is being blocked (Fig. 6a & S6). Excitingly, our functional analysis demonstrates that Hh instructs FAPs to produce Timp3 to control adipogenesis (Fig. 7c) and Gdf10 to regulate myogenesis (Fig. 7d).

3) Also, mono-cultures of FAPs isolated from the experimental mouse models and conditions should be used to independently determine FAPs phenotype *ex vivo* after cultures

We thank the reviewer for this important suggestion. We have now added new *in vivo* data demonstrating that loss of Dhh does not impact FAP expansion or proliferation but rather results in increased adipogenic differentiation, ultimately leading to increased IMAT deposition as observed in our Dhh null mice (Fig. 1e&f). Interestingly, when we analyzed what happens to FAPs in FAP-no-Ptch1 mice, we found that sustained Hh activation causes increased cell death resulting in fewer FAPs (Fig. 5c&d). The FAPs that remain are then forced to adopt a myofibroblast instead of an adipogenic fate causing massive scar tissue formation (Fig. 5b). Together, our new data demonstrate that endogenous Hh signaling acts as a potent adipogenic differentiation block, while too much Hh signaling causes cell death and a fate switch.

Reviewer #3: In this paper the authors identify the hedgehog family member that is involved in muscle regeneration, confirm its effects on interstitial fat deposition and expand their analysis of this pathway to show a defect in myofiber regeneration. They further identify a downstream mediator on the effect on adipogenicity, and they show that it does not mediate the effect on fiber regeneration. This is an interesting and informative piece of work, and while much could be added to clarify a number of arising questions, it already contains enough findings to be a significant contribution to the field. I was particularly impressed by the fact that the authors managed to separate the adipogenic phenotype from the defect in myofiber regeneration, which is a classical chicken-and-egg problem in the field.

We were grateful to read that “This is an interesting and informative piece of work.” We are also hopeful that our revised manuscript containing multiple new data points will “clarify a number of arising questions”, despite the reviewer arguing that “it already contains enough findings to be a significant contribution to the field.”

The results presented in figure one are apparently compelling, but more information should be provided: If the Umap plots presented are the results of aggregating a number of publicly available dataset, how many of the datasets are contributing to the results displayed? As an example, it is evident that Gli1,2 and 3 are expressed in a small minority of the FAPs and myogenic cells in figure 1c. Is this minority originating from one of the aggregated datasets, or from multiple? In particular Gli3 in FAP{s appears to follow an alternating patterns, with higher frequency of expressing cells (albeit lower absolute levels of expression) on days 1, 3.5 and 10, while showing a different patterns (higher expression in rare cells) in the rest of the time course. Is it because the data from days are contributed by a specific set of sequencing samples? This needs to be clarified

We apologize for not providing enough information. To address this concern, we have now amended the corresponding method section accordingly (“...and results displayed as aggregated data from all datasets”) and included links to our github page, that contains a detailed explanation and all the code used to prepare and analyze the data (<https://github.com/mckellardw/scMuscle>), and to a Dryad page, from which one can download our fully preprocessed data ([doi:10.5061/dryad.t4b8qtj34](https://doi.org/10.5061/dryad.t4b8qtj34)).

Briefly, we filtered the aggregated data to remove injury timepoints which did not have more than 5,000 cells to reduce the noise in the resulting analysis. Of the 111 samples in the original dataset, 94 remained

with an average of ~3,172 cells per sample after this filtering. In this dataset, Gli1 is detected in 2,813 cells from 86 samples, Gli2 is detected in 4,279 cells from 87 samples, and Gli3 is detected in 15,759 cells from 91 samples. Generally, the patterns observed here are caused by different numbers of cells in each of the injury timepoints. It is important to note that lowly expressed transcripts, including Gli1, Gli2, and Gli3, may appear sparsely expressed due to sampling biases, also known as dropout effects, in single-cell RNA-sequencing data. While it may appear that these genes are only expressed in a small number of cells, it is likely that because the transcripts are not very abundant, they are not detected for every cell in which they are truly expressed.

Thus, as the low abundance of the Glis could skew the interpretation of the presented data, we are now only focusing on the UMAP plots and have removed the expression values for the individual time points (Fig. 4a). This allows us to bring across the main point that of all cell populations present in muscle FAPs and MuSCs express crucial Hh pathway components. We then present new data showing that, while MuSCs may respond to Hh, they do so very weakly compared to FAPs (Fig. 4b). Next, we demonstrate that this weak Hh activity in MuSCs has no impact on myogenesis, further solidifying our data that Hh controls myogenesis indirectly through FAPs (Fig. 4c-d).

The results in figure 5, in which the authors explore the timing of activation of the pathway, are interesting but not too surprising, as it seemed unlikely that DHH would act after the initial wave of apoptosis that reduces FAPs numbers following acute damage. Still, it is interesting to observe that the positive effects of this pathway's activation on fibers is also taking place at about the same time, as this supports the proposal that the effect on myofibers is also mediated by FAPs. It would be great if the authors could include in the text a statement on the pharmacokinetics of the smoothed agonist in vivo, ideally in mice, as that would help interpreting the results.

We appreciate this insightful comment. Unfortunately, we are not aware of any study that determines the PK/PD properties of SAG. In fact, the lack of careful timing and dosing data of how SAG might influence muscle regeneration, was a major motivator for these experiments. To add more biological clarity on the PK/PD properties of SAG, we injured mice and administered SAG at 2 days after injury, the only time point that displayed benefits, and assessed Hh activation 6- and 12 hrs post injection (Fig. S3a). Interestingly, we find that SAG rapidly induces Hh indicating that FAPs are primed and ready to not only receive the signal but also convert this into a transcriptional response within 6-12 hrs.

The results presented in figure 6/7 are the most exciting of the manuscript, suggesting that DHH indeed works through FAPs. Indeed the defect in regeneration in animals in which FAPs lack *ptch* is impressive however in order to conclude that this is due to a differentiation block, the authors should really exclude an effect on survival or expansion. This is particularly important given that Pax7 cells did not change in numbers or BrdU positivity. BrdU should be shown for MyoD positive and ideally MyoG positive cells as well (less important as the drop in numbrs takes place between the Pax7 and myoD stages). Given the significant drop in MyoD positive cell numbers, and the lack of differences in the proliferation of Pax7 cells, what happened to the "missing" cells? A differentiation block without changes in proliferation or survival should lead to an accumulation of cells at a more immature stage.

We appreciate the concern about the "missing" cells. After careful re-analysis of the raw images and their quantifications by multiple members of the lab, I regretfully have to report a major mistake that happened during the Pax7 data analysis, which resulted in scrambling of not only the genotypes but also their respective values. After consulting with my former technician, who collected the data, he told me that he imaged the 3 and 5 dpi cohorts over multiple days. During unblinding, he, unfortunately, mixed up the value-to-genotype data between the time points, completely randomizing the data. I am incredibly sorry and embarrassed for this mistake. New procedures have been put in place now for this to never happen again.

To rectify this mistake, and to ensure correct analysis of the fate of MuSCs in FAP-no Ptch1 mice, we have re-genotyped every single sample from the 3dpi cohort and re-quantified the data. Our new data indicate now that FAP-specific Hh activation results does block the proliferation and expansion of Pax7+ MuSCs at 3dpi (Fig. 6b). We have also generated a brand new and independent 5 dpi FAP-no-Ptch1 cohort. With this cohort, we were also able to administer BrdU, which allowed us to also look at the proliferation rates of Pax7+ & MyoD+ cells at day 5. Our new results demonstrate that fewer Pax7+ MuSCs and MyoD+ myoblasts exist at 5 dpi and that both also display reduced proliferation (Fig. 6b&c). Importantly, our new quantifications of the % of MyoD+ cells are very similar to our old ones (see picture). Thus, ectopic Hh activation blocks early MuSCs expansion resulting in a reduced myoblast pool, and subsequently in smaller myofibers that contain fewer myonuclei.

I was also very surprised that the effect of knocking out ptch on FAP numbers, proliferation and survival was not addressed at all... it seems like an important parameter.

We agree with the reviewer and have now assessed the total number, proliferation rate and survivability of FAPs in FAP-no-Ptch1 mice. We find that sustained high-level Hh activation increases cell death, with no effect on proliferation, resulting in reduced number of FAPs (Fig. 5c-d). We also show that the remaining FAPs are pushed away from an adipogenic towards a fibrogenic fate (Fig. 5b). We also analyzed FAP fate in Dhh null animals, and found that, while loss of Hh signaling has no impact on FAP numbers or proliferation, the differentiation of FAPs into adipocytes is increased (Fig. 1e&f). These data strongly support our findings that Hh, through Dhh, acts as endogenous adipogenic brake.

Finally, it is not clear to me why the authors looked at the myogenic defect only with glycerol damage... is this effect also present when CTX is used?

We agree with the reviewer and have now extended our analysis of the FAP-no-Ptch1 model to also include a CTX injury. Similarly to the GLY injury and our SAG CTX experiments, 21 days post CTX FAP-no-Ptch1 mice display reduced IMAT, smaller CSA of myofibers and increased fibrosis (Fig. S4b). Thus, sustained high Hh activation either globally or FAP-specifically blocks myogenesis (indirectly through FAPs) and forces FAPs away from an adipogenic and towards a fibrotic fate.

Thank you again for your extensive help with this manuscript. I appreciate your efforts and suggestions, which have substantially improved our work. Please let me know if I can provide any additional information or answer any further questions.

REVIEWERS' COMMENTS

Reviewer #1 (Remarks to the Author):

In this revised manuscript by Norris et al., the authors performed extensive new experiments to address the mechanism linking hedgehog-activated repression of adipogenesis with muscle regeneration. The authors provided new data to show that muscle stem cells themselves do not respond to Hh activation, which acts in a non-cell autonomous fashion to modify the ability of FAPs to provide trophic factors for MuSC differentiation and fusion. This is an elegant model that nicely explains the requirement of DHH to simultaneously suppress adipogenesis and promote muscle repair through FAPs. These revisions merit publication as it demonstrates the multifaceted roles of hedgehog signaling in modifying different components in the muscle stem cell niche, with very nice data presented to support their model.

My only minor concerns/suggestions for improvement are regarding data interpretation. One confusing aspect of the paper for the general audience is that while hedgehog activation appears necessary for muscle regeneration, it also appears to be deleterious for muscle regeneration in certain experimental settings. The author has done a nice job to show that constitutive Hh activation actually leads to FAP apoptosis and reduction of FAPs overall. This data would explain why overactivation of Hh also leads to defect in muscle regeneration, suggesting that Hh has a cell-autonomous role in pruning the FAP population. However, schematic diagrams like Fig. 6E and 7E seems to contradict each other. FAP Hh activation is suppressing muscle formation in Fig. 6E, contrary to the diagram presented in Fig. 7E that shows Hh activation of FAPs promotes muscle formation through GDF10. I hope that the authors can see how that can be confusing to the readers. I might add something to the summary model in Figure 8 to show how over-activation of Hh can also lead to reduction of FAPs that could adversely impact muscle regeneration, and differentiate between physiologic/dynamic Hh activation (reparative, ample GDF10+ FAPs) vs. constitutive/non-physiologic Hh activation (maladaptive, reduction of GDF10+ FAPs).

My other minor concern is the interpretation of the data in Fig 5 that “ectopic Hh activation pushes FAPs to adopt a myofibroblast fate.” While that is likely to be true, I do not believe the authors can definitively claim that without lineage tracing studies (tagging FAPs with fluorescent markers prior to injury).

Reviewer #2 (Remarks to the Author):

The authors have been responsive to the reviewer comments, by adding important data in support to their conclusions. The finding that Hh restricts IMAT formation and promotes muscle regeneration post injury, through its ligand DHH, is a substantial extension of their previous works and is now also supported by mechanistic insights into the mediators of FAP/MuSC interplay. Overall, in the opinion of this reviewer these findings are worth of publication in Nat Comm

Reviewer #1: *In this revised manuscript by Norris et al., the authors performed extensive new experiments to address the mechanism linking hedgehog-activated repression of adipogenesis with muscle regeneration. The authors provided new data to show that muscle stem cells themselves do not respond to Hh activation, which acts in a non-cell autonomous fashion to modify the ability of FAPs to provide trophic factors for MuSC differentiation and fusion. This is an elegant model that nicely explains the requirement of DHH to simultaneously suppress adipogenesis and promote muscle repair through FAPs. These revisions merit publication as it demonstrates the multifaceted roles of hedgehog signaling in modifying different components in the muscle stem cell niche, with very nice data presented to support their model.*

We greatly appreciate the reviewer's kind words. The previous comments and suggestions were a fantastic guide to vastly improve our manuscript. Thus, Thank you!

My only minor concerns/suggestions for improvement are regarding data interpretation. One confusing aspect of the paper for the general audience is that while hedgehog activation appears necessary for muscle regeneration, it also appears to be deleterious for muscle regeneration in certain experimental settings. The author has done a nice job to show that constitutive Hh activation actually leads to FAP apoptosis and reduction of FAPs overall. This data would explain why overactivation of Hh also leads to defect in muscle regeneration, suggesting that Hh has a cell-autonomous role in pruning the FAP population. However, schematic diagrams like Fig. 6E and 7E seems to contradict each other. FAP Hh activation is suppressing muscle formation in Fig. 6E, contrary to the diagram presented in Fig. 7E that shows Hh activation of FAPs promotes muscle formation through GDF10. I hope that the authors can see how that can be confusing to the readers. I might add something to the summary model in Figure 8 to show how over-activation of Hh can also lead to reduction of FAPs that could adversely impact muscle regeneration, and differentiate between physiologic/dynamic Hh activation (reparative, ample GDF10+ FAPs) vs. constitutive/non-physiologic Hh activation (maladaptive, reduction of GDF10+ FAPs).

We thank the reviewer for this valid comment. In order to avoid confusion, we have edited the summary models in Figs. 2-3 and 5-7 and the text throughout to better distinguish "ectopic" from "endogenous" Hh activation. We have also modified our main model (Fig 8) and included an additional schematic that distinguishes "physiological Hh activation" through "endogenous" DHH from "ectopic Hh activation".

My other minor concern is the interpretation of the data in Fig 5 that "ectopic Hh activation pushes FAPs to adopt a myofibroblast fate." While that is likely to be true, I do not believe the authors can definitively claim that without lineage tracing studies (tagging FAPs with fluorescent markers prior to injury).

We agree with the reviewer and apologize for overinterpretation of our data. We have now edited the ms correspondingly including suggesting using genetic lineage tracing to determine if FAPs indeed turn into myofibroblasts instead of adipocytes (see discussion).

Reviewer #2: *The authors have been responsive to the reviewer comments, by adding important data in support to their conclusions. The finding that Hh restricts IMAT formation and promotes muscle regeneration post injury, through its ligand DHH, is a substantial extension of their previous works and is now also supported by mechanistic insights into the mediators of FAP/MuSC interplay. Overall, in the opinion of this reviewer these findings are worth of publication in Nat Comm.*

We thank the reviewer for the kind words and all the previous helpful comments.

Again, I extend my deep thanks for your extremely helpful insights into this work. Your recommendations and suggestions have improved the work substantially beyond the original. Please let me know if I can provide any additional information or answer any further questions